# Single-pass transformation of syngas into ethanol with high selectivity by triple tandem catalysis

Jincan Kang[1,2], Shun He[1,2], Wei Zhou[1,2], Zheng Shen[1], Yangyang Li[1], Mingshu Chen [1], Qinghong Zhang[1✉] & Ye Wang [1✉]

Synthesis of ethanol from non-petroleum carbon resources via syngas (a mixture of $H_2$ and CO) is an important but challenging research target. The current conversion of syngas to ethanol suffers from low selectivity or multiple processes with high energy consumption. Here, we report a high-selective conversion of syngas into ethanol by a triple tandem catalysis. An efficient trifunctional tandem system composed of potassium-modified ZnO–ZrO$_2$, modified zeolite mordenite and Pt–Sn/SiC working compatibly in syngas stream in one reactor can afford ethanol with a selectivity of 90%. We demonstrate that the $K^+$–ZnO–ZrO$_2$ catalyses syngas conversion to methanol and the mordenite with eight-membered ring channels functions for methanol carbonylation to acetic acid, which is then hydrogenated to ethanol over the Pt–Sn/SiC catalyst. The present work offers an effective methodology leading to high selective conversion by decoupling a single-catalyst-based complicated and uncontrollable reaction into well-controlled multi-steps in tandem in one reactor.

[1] State Key Laboratory of Physical Chemistry of Solid Surfaces, Collaborative Innovation Center of Chemistry for Energy Materials, National Engineering Laboratory for Green Chemical Productions of Alcohols, Ethers and Esters, College of Chemistry and Chemical Engineering, Xiamen University, Xiamen 361005, China. [2]These authors contributed equally: Jincan Kang, Shun He, Wei Zhou. ✉email: zhangqh@xmu.edu.cn; wangye@xmu.edu.cn

Syngas ($H_2$/CO), which can be produced from natural (shale) gas, coal, biomass, carbon-containing waste and carbon dioxide, is one of the most important platforms for utilization of non-petroleum resources to supply energy and chemical feedstocks. The depletion of crude oil and the growing demand for fuels and chemicals have stimulated recent research activities in syngas transformations[1–3]. As an ideal fuel additive, a promising hydrogen carrier and a versatile building-block for chemicals, fuels and polymers[4,5], ethanol is one of the most attractive molecules that may be produced from syngas. The current production of ethanol relies on the fermentation of sugars, but this process suffers from the competition with food supply, the high energy consumption in product separation and purification, and the limited efficiency and ethanol selectivity[6–8]. The chemical synthesis of ethanol from various carbon resources via syngas represents one of the most promising approaches.

Many studies have been devoted to the direct conversion of syngas into ethanol with a single catalyst (Fig. 1, Route A)[5,8–11]. This reaction is complicated because of many elementary steps involved, which include $H_2$ dissociation, CO dissociative and non-dissociative activation, formation of adsorbed CO, $CH_xO$ and $CH_x$ intermediates, C–C coupling between different intermediates, and formation of products[5,8,9,11]. The multiple intermediates lead to different reaction channels, forming methane, methanol, $C_{2+}$ hydrocarbons and $C_{2+}$ oxygenates together with ethanol. The selective synthesis of ethanol requires the precise control of C–C coupling between intermediates from CO dissociative and non-dissociative activation. The design of a single catalyst with different functionalities to enable only a single reaction channel is very difficult. Although much effort has been devoted to designing bi- or multi-component catalysts such as Rh–Mn, Rh–Fe, Cu–Co and Cu–Fe catalysts, the ethanol selectivity is lower than 60% even at limited CO conversions (Supplementary Table 1)[12–19].

Syngas can also be transformed into ethanol through indirect routes via methanol or dimethyl ether (DME) synthesis, followed by carbonylation with CO and subsequent hydrogenation of acetic acid or methyl acetate (Fig. 1, Routes B and C). The indirect routes suffer from multiple processes and energy-consuming product separation/purification in each process. The recently developed route via DME and methyl acetate (Fig. 1, Route C) is more cost-efficient and environmental friendly because of the use of halogen-free heterogeneous H-form mordenite (H-MOR) zeolite to replace iodide-promoted Rh complex homogeneous catalyst for carbonylation[20,21]. However, the overall selectivity of ethanol is limited to 67% due to methanol formation in subsequent hydrogenation. The development of new routes for single-pass conversion of syngas to ethanol with high selectivity is very attractive but highly challenging.

Tandem catalysis, which can perform multi-step sequential reactions in a single reactor or one pot, has offered an opportunity to discover new catalytic processes with better efficiencies[22–25]. The conversion of syngas to $C_2$–$C_4$ olefins or aromatics beyond traditional Fischer-Tropsch synthesis has recently been achieved by designing bifunctional catalysts that work in tandem for CO hydrogenation to $CH_3OH$/DME or ketene intermediate and conversion of intermediate to target product[26–30]. Unlike uncontrollable chain growth on Fischer-Tropsch metal surfaces, the C–C coupling in these systems occurs inside zeolite cages, where the confinement effect narrows the product distribution by shape selectivity. However, the high-selective synthesis of a specific product, in particular a $C_{2+}$ oxygenate, by using the concept of shape selectivity is very challenging.

The decoupling of a complicated reaction with many reaction channels into controllable multi-steps would enable the control of reaction channel. Unlike the bifunctional catalysis using the shape selectivity of zeolite to control the product selectivity[26–30], the present work aims at reaction channel-controlled C–C coupling. Our idea is that the syngas reactant enables us to perform not only hydrogenation but also carbonylation, which produces a $C_2$ oxygenate. A few recent studies have adopted this methodology for the synthesis of $C_2$ oxygenates. We succeeded in synthesizing methyl acetate from syngas by tandem catalysis via DME intermediate[31]. Although ethanol was obtained by using a three-component catalytic system composed of layer-by-layer $ZnAl_2O_4$, H-MOR and $ZnAl_2O_4$, the selectivity of ethanol was only ~50%. A sandwich-configuration catalytic system composed of $CuZnO_x$, CuH-MOR and $CuZnO_x$ was also reported for syngas conversion, but the ethanol selectivity was only 15%[32]. The co-feeding of DME and syngas over dual-bed catalysts for DME carbonylation and methyl acetate hydrogenation provided ethanol with selectivity of <50%[33–35].

Here, we report the design of new tandem catalytic systems with high ethanol selectivity (Fig. 1, Route D). We discovered that the integration of methanol synthesis, methanol carbonylation and acetic acid hydrogenation in tandem in one reactor could accomplish the selective conversion of syngas into ethanol with selectivity as high as 90%. We demonstrate that the careful design of each catalyst to direct the reaction channel to the target intermediate or product, the interplay between the three steps and the compatibility of each catalyst in syngas stream are crucial to ethanol selectivity. The present work opens an avenue for the innovation of selective conversion route by decoupling an uncontrollable reaction, which has multiple elementary steps and

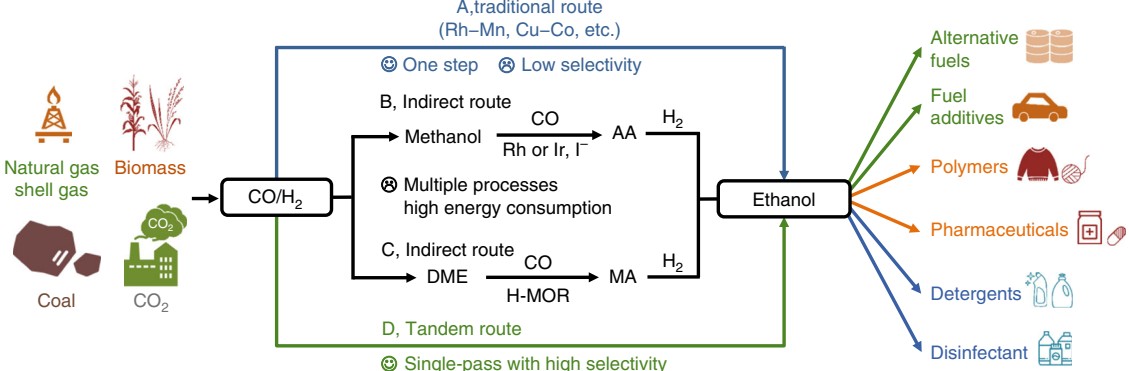

**Fig. 1 Routes for conversion of syngas to ethanol.** Route A, traditional and direct route based on a single catalyst. Route B, indirect route including three processes, i.e., methanol synthesis, methanol carbonylation with CO and acetic acid (AA) hydrogenation. Route C, indirect route including three processes, i.e., syngas to dimethyl ether (DME), DME carbonylation with CO and methyl acetate (MA) hydrogenation. Route D, Tandem catalytic route of this work.

many reaction channels, into well-controlled multi-steps with tandem catalysis.

## Results

**Composition and performance of tandem catalysts.** Thermodynamic analyses reveal that all the three tandem steps, i.e., methanol synthesis, methanol carbonylation and acetic acid hydrogenation, are thermodynamically more feasible at lower temperatures (Supplementary Fig. 1). A higher temperature is particularly unfavourable to methanol synthesis. However, methanol carbonylation and acetic acid hydrogenation usually need to proceed at 500–600 K because of the kinetic requirement[36,37]. CO equilibrium conversions for ethanol synthesis via tandem catalysis were evaluated to be 38% and 12%

at 550 and 600 K, respectively, under a syngas ($H_2/CO = 1$) pressure of 5.0 MPa (Supplementary Fig. 1 and Supplementary Note 1). To match methanol carbonylation and acetic acid hydrogenation reactions, we first investigated catalytic behaviours of some typical metal oxides that are capable of catalysing syngas conversion to $CH_3OH/DME$ at 500–600 K. $ZnAl_2O_4$, $ZnGa_2O_4$ and $ZnCr_2O_4$ compounds with spinel structure were selective for DME formation (Fig. 2a). On the other hand, $ZnO–ZrO_2$ ($Zn/Zr = 1:8$) and $ZrO_2–In_2O_3$ ($Zr/In = 1:2$) composites showed higher $CH_3OH$ selectivity, but the formation of undesirable $CH_4$ was serious over the $ZrO_2–In_2O_3$ possibly because of its high hydrogenation ability[38]. The modification of the $ZnO–ZrO_2$ by $K^+$ ($K/Zn/Zr = 0.07:1:8$) further improved $CH_3OH$ selectivity to 93% (Fig. 2a). Over these metal oxides, $CO_2$ was also formed, indicating the occurrence of water-gas shift (WGS) reaction caused by

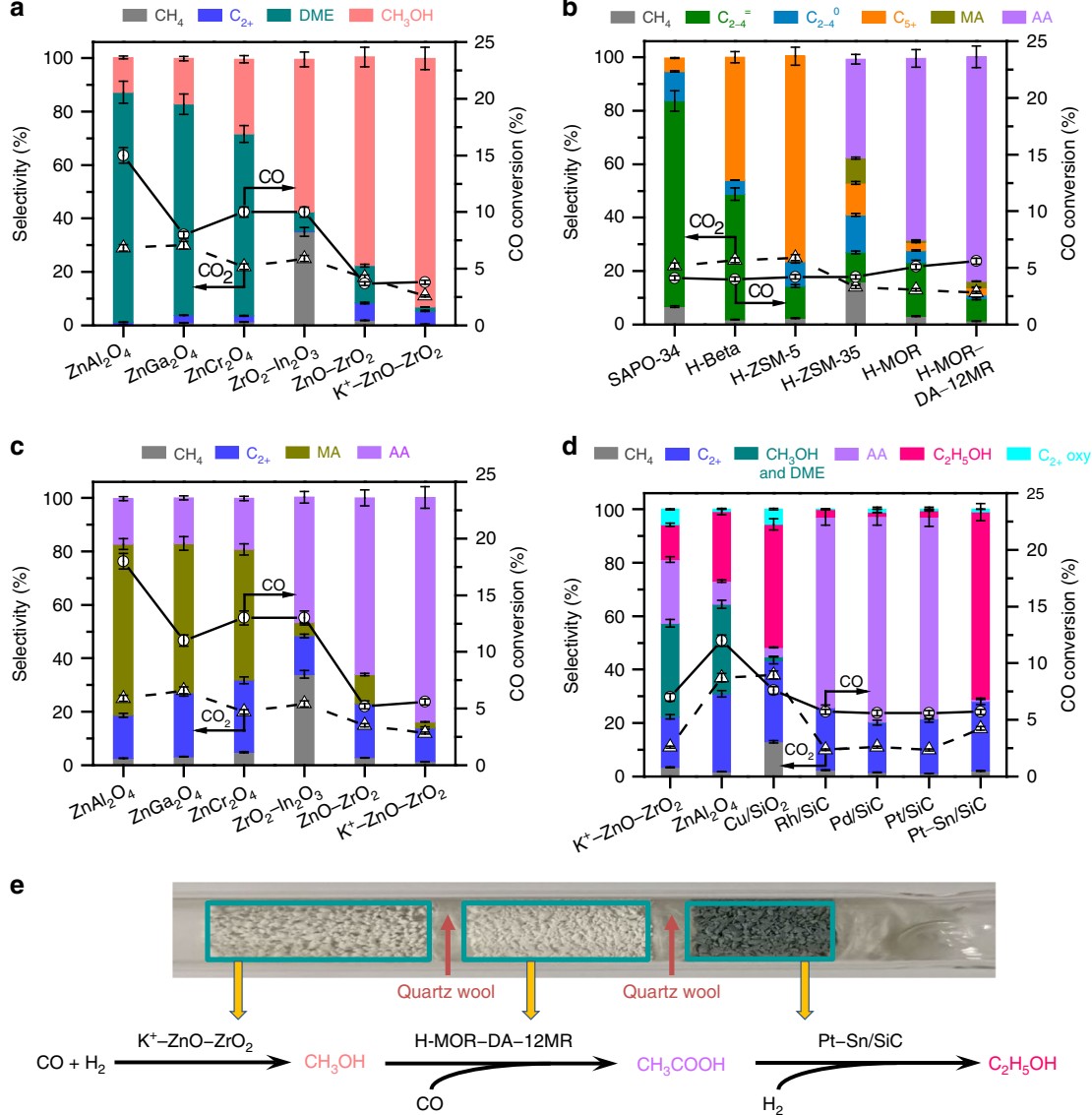

**Fig. 2 Catalytic behaviours and reaction pathways. a** Metal oxides alone. **b** Combinations of $K^+–ZnO–ZrO_2$ and zeolites. **c** Combinations of metal oxides and H-MOR-DA-12MR. **d** Combinations of $K^+–ZnO–ZrO_2$, H-MOR-DA-12MR and hydrogenation catalysts. **e** Reaction pathways for direct synthesis of ethanol from syngas. $C_{2+}$: $C_{2+}$ hydrocarbons; DME: dimethyl ether; $C_{2-4}^=$: $C_2$–$C_4$ olefins; $C_{2-4}^0$: $C_2$–$C_4$ paraffins; $C_{5+}$: $C_{5+}$ hydrocarbons; MA: methyl acetate; AA: acetic acid; $C_{2+}$ oxy.: ethyl acetate and methyl acetate. Reaction conditions: weights of metal oxide, zeolite and hydrogenation catalyst = 0.66, 0.66 and 0.66 g; $H_2/CO = 1:1$; $P = 5.0$ MPa; $T = 583$ K; $F = 25$ mL min$^{-1}$; time on stream, 20 h. The selectivity was calculated on a molar carbon basis. Carbon balances were 95–99%. The experiments in each case were performed for three times. The error bar represents the relative deviation, which is within 5%.

$H_2O$ formed during syngas conversions. The $CO_2$ selectivity, which was calculated separately from the hydrogenation products, decreased in the sequence of $ZnGa_2O_4 > ZnAl_2O_4 > ZrO_2–In_2O_3 > ZnCr_2O_4 > ZnO–ZrO_2 > K^+–ZnO–ZrO_2$.

The integration of the $K^+–ZnO–ZrO_2$ and a zeolite in tandem separated by quartz wool (denoted as $K^+–ZnO–ZrO_2 │ zeolite$) resulted in different products depending on zeolite topology (Fig. 2b). Lower olefins were formed with a selectivity of ~80% using $K^+–ZnO–ZrO_2 │ SAPO-34$ combination, in agreement with the fact that SAPO-34 is an excellent methanol-to-olefins (MTO) catalyst[39]. Higher hydrocarbons with carbon numbers of ≥ 5 ($C_{5+}$) were formed with considerable selectivity using $K^+–ZnO–ZrO_2 │ H-beta$ and $K^+–ZnO–ZrO_2 │ H-ZSM-5$, also coinciding with the effect of zeolite topology on product distribution in methanol-to-hydrocarbons (MTH) reaction[40]. Uniquely, besides hydrocarbons, acetic acid was formed by combining H-ZSM-35 or H-MOR with the $K^+–ZnO–ZrO_2$ (Fig. 2b). It is known that H-MOR and H-ZSM-35, both of which possess eight-membered-ring (8-MR) channels, are capable of catalysing the carbonylation of DME with CO to methyl acetate at 423–473 K[21,41,42]. The Brønsted acid sites located inside the 8-MR channels are proposed for carbonylation through the Koch-type reaction, while those inside the 12-MR or 10-MR channels are responsible for the formation of hydrocarbons and coke[21,41–43]. Although H-MOR is known to be also capable of catalysing the carbonylation of methanol at 500–600 K[36], the catalyst requirement for methanol carbonylation is less understood. We found that the selective removal of framework Al in the 12-MR channels (denoted as H-MOR–DA–12MR) increased acetic acid selectivity to 84% (Fig. 2b). The combination of $K^+–ZnO–ZrO_2 │ H-MOR–DA–12MR$ showed the lowest $CO_2$ selectivity. When the $ZnAl_2O_4$, $ZnGa_2O_4$ or $ZnCr_2O_4$, which catalysed DME formation, was integrated with the H-MOR–DA–12MR, methyl acetate was formed as the major product (Fig. 2c), indicating the occurrence of DME carbonylation over the H-MOR–DA–12MR. The change in $CO_2$ selectivity with metal oxide for the metal oxide │ H-MOR–DA–12MR combination showed a similar trend to that for metal oxides alone. The comparison of Figs. 2a and c demonstrates that the final product can be directed by the intermediate in tandem system; $CH_3OH$ as an intermediate leads to acetic acid, while DME results in methyl acetate. Rh-based catalysts as well as an indirect route integrated by multiple reaction processes via methanol, DME and methyl acetate have been exploited for the conversion of syngas into acetic acid (Supplementary Table 2). The present $K^+–ZnO–ZrO_2 │ H-MOR–DA–12MR$ system shows significantly higher acetic acid selectivity than most of the Rh-based catalysts or the multi-process route.

The triple tandem system composed of $K^+–ZnO–ZrO_2 │ H-MOR–DA–12MR$ and a hydrogenation catalyst was designed for the conversion of syngas to ethanol. The use of metal oxides, i.e., $K^+–ZnO–ZrO_2$ and $ZnAl_2O_4$, as the hydrogenation catalyst in the downstream only showed low ethanol selectivity (Fig. 2d). Methanol and DME were also formed, suggesting that the metal oxide catalysed the further conversion of remaining syngas. The use of $Cu/SiO_2$ provided a higher ethanol selectivity (46%), but $CH_4$ was formed with a considerable selectivity (13%) at 583 K. The ethanol selectivity became lower at lower temperatures in the case of $Cu/SiO_2$ because of the decreased efficiency of acetic acid hydrogenation (Supplementary Table 3). It is unexpected that the Rh/SiC, Pd/SiC and Pt/SiC were almost inactive for the hydrogenation of acetic acid in the triple tandem system, although supported noble metals were reported to catalyse this reaction[37]. Interestingly, the modification of Pt/SiC by Sn completely changed the major product in the triple tandem system from acetic acid to ethanol (Fig. 2d). The ethanol

selectivity increased with the content of Sn in the Pt–Sn/SiC catalyst and was saturated at 70% at 583 K when the Sn content reached 1.2 wt% (Supplementary Table 4). $C_2$–$C_4$ olefins and $CO_2$ were the major by-products, and their selectivities were 16% and 18%, respectively. The increase of Pt loading in the Pt–Sn/SiC from 0.5 to 1.0 wt% slightly increased ethanol selectivity, but a higher Pt loading (>1.2 wt%) would promote the WGS reaction to form more $CO_2$ and the hydrogenation of lower olefins to paraffins (Supplementary Table 5). The results in Fig. 2a–d clearly demonstrate that the conversion of syngas to ethanol proceeds through a tandem mechanism via methanol and acetic acid intermediates (Fig. 2e).

The catalytic performance depends on the ratio of amounts of the three catalyst components. The increase in the amount of $K^+–ZnO–ZrO_2$ from 0.66 to 1.00 g in the $K^+–ZnO–ZrO_2 │ H-MOR–DA–12MR │ Pt–Sn/SiC$ increased CO conversion from 5.7 to 8.4% and the ethanol selectivity decreased only slightly to 68% at 583 K (Supplementary Table 6). A further increase in the amount of $K^+–ZnO–ZrO_2$ to 1.50 g significantly decreased the ethanol selectivity. The increase in the amount of H-MOR–DA–12MR slightly improved the CO conversion and ethanol selectivity. The amount of Pt–Sn/SiC did not affect CO conversion, but a sufficient amount of Pt–Sn/SiC was required to keep ethanol selectivity high. We obtained a 9.7% CO conversion at ethanol selectivity of 64%.

**Structural characterizations of tandem catalysts.** X-ray diffraction (XRD) measurements showed that only tetragonal $ZrO_2$ and cubic $In_2O_3$ crystalline phases were observed in the $ZnO–ZrO_2$ (Zn/Zr = 1:8) and $ZrO_2–In_2O_3$ (Zr/In = 1:2) catalysts, respectively, whereas spinel-structured compounds were formed for the $ZnAl_2O_4$, $ZnGa_2O_4$ and $ZnCr_2O_4$ (Supplementary Fig. 2a). Tetragonal $ZrO_2$ was also the sole phase in the $K^+–ZnO–ZrO_2$ catalyst. The mean size of $K^+–ZnO–ZrO_2$ particles estimated from transmission electron microscopy (TEM) was 9.1 nm (Fig. 3a). High-resolution TEM (HRTEM) micrographs for this sample showed the lattice fringes that could only be attributed to tetragonal $ZrO_2$ {011} facet (Fig. 3b). These results suggest that ZnO was highly dispersed inside the lattice of $ZrO_2$ to form solid solution, and this phenomenon was also reported previously[30,44].

Ammonia temperature-programmed desorption ($NH_3$-TPD) indicated the existence of acid sites on our metal oxides (Supplementary Fig. 2b). The density of acid sites decreased in the order of $ZnAl_2O_4 > ZnGa_2O_4 > ZnCr_2O_4 > ZnO–ZrO_2 > ZrO_2–In_2O_3 > K^+–ZnO–ZrO_2$. Pyridine-adsorbed Fourier-transform infrared (FT-IR) measurements suggested that Lewis acid sites existed on these metal oxides (Supplementary Fig. 2c) and the density of Lewis acid sites decreased in a trend similar to that from $NH_3$-TPD. This agrees well with the order of increase in $CH_3OH$ selectivity and that of decrease in DME selectivity (Fig. 2a). Thus, the acidity of metal oxides influences the product selectivity in syngas conversions by catalysing the dehydration of $CH_3OH$ to DME. The modification of $ZnO–ZrO_2$ with $K^+$ decreases the acidity, thus improving $CH_3OH$ selectivity.

Oxygen vacancy sites on $ZrO_2$ surfaces have been proposed to account for CO activation[45]. Our electron paramagnetic resonance (EPR) spectroscopy measurements showed a signal at $g$ value of 2.003 (Supplementary Fig. 3), which could be attributed to oxygen vacancies over $ZrO_2$-based metal oxides[46]. The incorporation of ZnO into $ZrO_2$ to form solid solution enhanced the signal at $g$ value of 2.003 and thus the generation of oxygen vacancies over both $ZnO–ZrO_2$ and $K^+–ZnO–ZrO_2$ samples. We believe that the increased density of oxygen vacancy sites favours the activation of CO as well as $H_2$[30,45,47].

The H-MOR–DA–12MR has been characterized by $^{23}Na$, $^{27}Al$ and $^1H$ magic spinning nuclear magnetic resonance (MAS NMR)

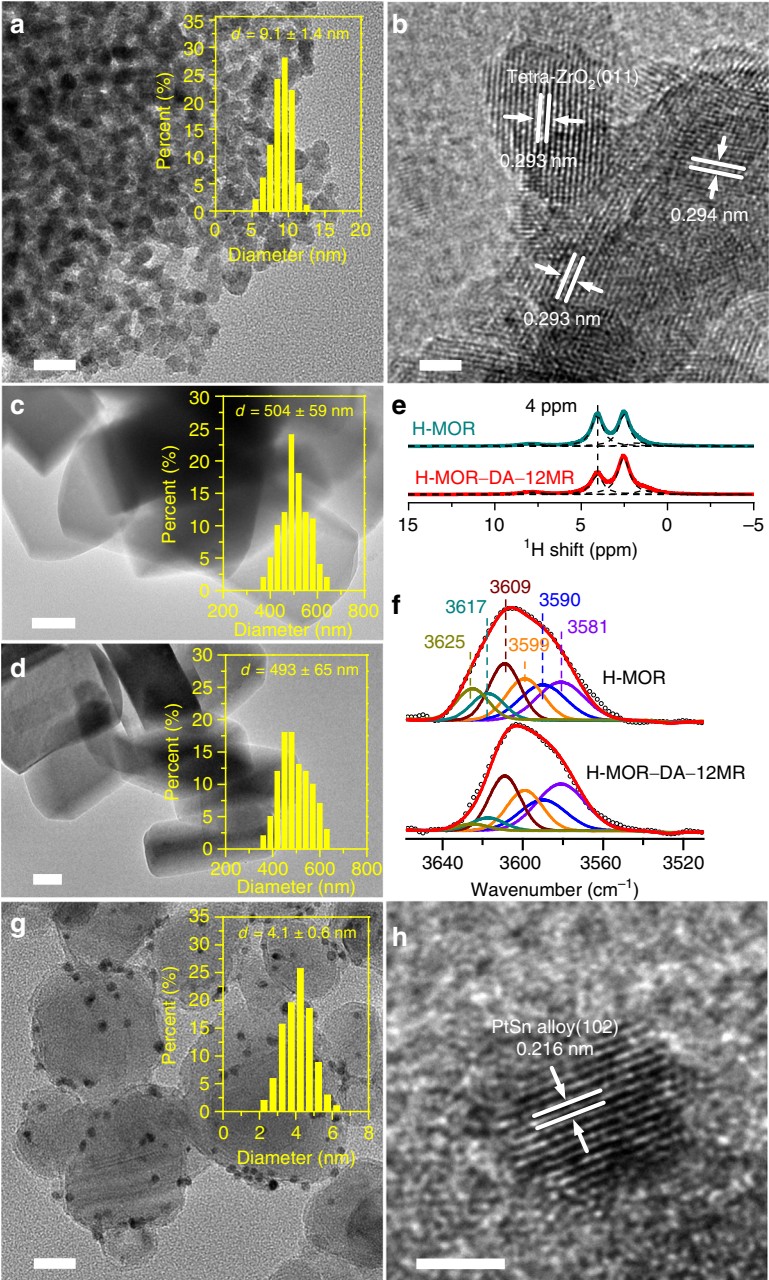

**Fig. 3 Characterization of tandem catalysts. a** TEM micrograph for $K^+$–ZnO–$ZrO_2$ particles with particle size distribution. Scale bar: 20 nm. **b** HRTEM micrograph for $K^+$–ZnO–$ZrO_2$ particles. Scale bar: 2 nm. **c** TEM micrograph for H-MOR with particle size distribution. Scale bar: 100 nm. **d** TEM micrograph for H-MOR–DA–12MR with particle size distribution. Scale bar: 100 nm. **e** $^1H$ MAS NMR for H-MOR and H-MOR–DA–12MR zeolites. The peak at 4.0 ppm can be attributed to the Brønsted acid site. **f** FT-IR spectra of H-MOR and H-MOR–DA–12MR zeolites. Assignments of deconvoluted bands: 3625, 3617 and 3609 $cm^{-1}$: Brønsted acid sites in 12-MR; 3599 $cm^{-1}$: Brønsted acid sites in intersections between 8-MR and 12-MR; 3590 and 3581 $cm^{-1}$: Brønsted acid sites in 8-MR. **g** TEM micrograph for Pt–Sn/SiC with particle size distribution. Scale bar: 20 nm. **h** HRTEM micrograph for a Pt–Sn particle. Scale bar: 2 nm.

spectroscopic studies. This sample was prepared by exchanging of $Na^+$ into pyridine-adsorbed H-MOR, removal of pyridine at 873 K and subsequent steam treatment at 723 K, followed by ion-exchanging with $NH_4^+$ and calcination to remove $Na^+$ cations[43]. Both XRD (Supplementary Fig. 4a) and TEM (Figs. 3c and d) confirmed that the crystalline structure and morphology of H-MOR were not changed. The $Na^+$ cations are expected to be mainly located in the 8-MR channel, where pyridine cannot enter. This was confirmed by $^{23}Na$ MAS NMR measurements, which could discriminate $Na^+$ cations in different locations[48], and the result showed that $Na^+$ mainly remained in 8-MR channels after steam treatment at 723 K (Supplementary Fig. 5). The Al species

with $Na^+$ in the ion-exchanging position are more robust toward steam treatment, and thus the Al species in 12-MR channels are more prone to dealumination during steam treatment. $^{27}Al$ (Supplementary Fig. 6) and $^1H$ MAS NMR (Fig. 3e) results suggest the removal of framework Al species and Brønsted acid sites (Fig. 3e)[49,50]. Pyridine-adsorbed FT-IR studies showed significant decreases in intensities of IR bands ascribed to acid sites after steam treatment (Supplementary Fig. 4b), suggesting the occurrence of dealumination in 12-MR channels, because pyridine can only enter the 12-MR channel. FT-IR measurements in a range of 3540–3640 $cm^{-1}$ also confirmed the partial removal of OH groups related to Brønsted acid sites in 12-MR channels

(Fig. 3f). The combination of the $^1H$ MAS NMR result, which could provide information of the total amount of Brønsted acid sites, and the deconvolution result of IR bands belonging to OH groups allowed us to estimate the amounts of Brønsted acid sites in 8-MR and 12-MR channels as well as in the intersection between the two channels[51,52]. After dealumination, about 57% of Brønsted acid sites in 12-MR channels were removed, whereas 89% of Brønsted acid sites in 8-MR channels were retained (Supplementary Table 7). We believe that the selective removal of Brønsted acid sites in 12-MR channels is the main reason for the enhancement in acetic acid selectivity by using the H-MOR–DA–12MR (Fig. 2b).

TEM measurements for the Pt/SiC and Pt–Sn/SiC catalysts showed that Pt or Pt–Sn nanoparticles were uniformly dispersed on SiC support and the mean size of Pt particles decreased in the presence of Sn (Supplementary Fig. 7). The mean size of Pt particles was 4.1 nm in the Pt–Sn/SiC with a Sn content of 1.2 wt% (Fig. 3g). Line-scan energy-dispersive X-ray spectroscopy (EDS) (Supplementary Fig. 8) and HRTEM measurements (Fig. 3h and Supplementary Fig. 9) indicated the formation of Pt–Sn alloy. Quasi in situ X-ray photoelectron spectroscopy (XPS) studies showed that the binding energy of Pt $4f_{7/2}$ was 70.9–71.0 eV for the Pt/SiC catalyst after either $N_2$ pretreatment at 673 K or syngas ($H_2/CO = 1$) pretreatment at 583 K, indicating that Pt on this catalyst was in metallic state (Supplementary Fig. 10a)[53,54]. The binding energy of Pt $4f_{7/2}$ shifted to 71.4–71.5 eV for the Pt–Sn/

SiC catalyst, suggesting that $Pt^{\delta+}$ also existed in the presence of Sn. The broad Sn 3d spectrum for the Pt–Sn/SiC catalyst after either $N_2$ or syngas pretreatment could be assigned to cationic Sn species ($Sn^{2+}$ and/or $Sn^{4+}$) and a small fraction of $Sn^0$ (Supplementary Fig. 10b)[54]. The ability of CO chemisorption decreased significantly by modifying the Pt/SiC with Sn (Supplementary Fig. 10c).

**Reaction kinetics and stability.** We found that the increase in reaction temperature increased CO conversion but decreased ethanol selectivity for the $K^+$–ZnO–ZrO$_2$ | H-MOR–DA–12MR | Pt–Sn/SiC system (Fig. 4a). The ethanol selectivity decreased more steeply at > 583 K. This is because the selectivity of CH$_3$OH decreased and those of C$_{2+}$ hydrocarbons and CO$_2$ increased significantly at >583 K over the $K^+$–ZnO–ZrO$_2$ catalyst (Supplementary Fig. 11). The CO conversion over this catalyst did not increase significantly at high temperatures due to the thermodynamic limitation (Supplementary Fig. 1). The ethanol selectivity reached 90% and 81% at CO conversions of 0.7% and 4.0% at 503 and 543 K, respectively (Fig. 4a). The reaction with the triple tandem system was performed at 543 K for six times with catalyst from different batches, and the performance with CO conversion of 4.0 ± 0.2% and ethanol selectivity of 81 ± 4% could be well reproduced (Supplementary Fig. 12). To the best of our knowledge, such high ethanol selectivity has never been reported before in syngas conversions (Supplementary Table 1).

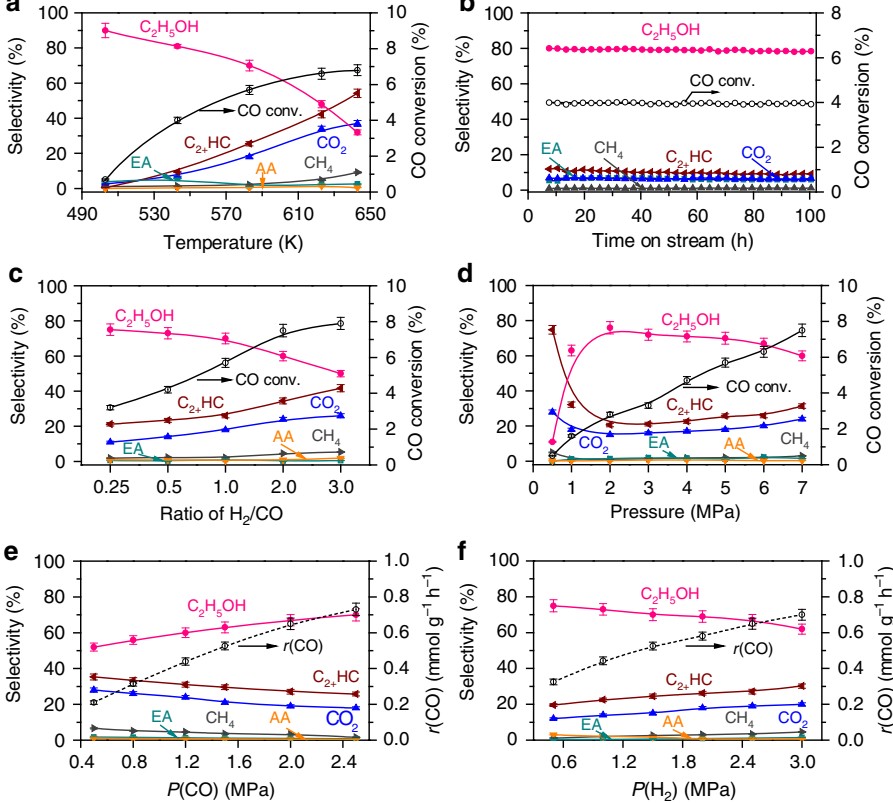

**Fig. 4 Reaction kinetics for the triple tandem system. a** Effect of reaction temperature. Reaction conditions: $H_2/CO = 1:1$; $P = 5.0$ MPa; $F = 25$ mL min$^{-1}$; time on stream, 20 h. **b** Catalyst stability. Reaction conditions: $H_2/CO = 1:1$; $P = 5.0$ MPa; $F = 25$ mL min$^{-1}$; $T = 543$ K. **c** Effect of $H_2/CO$ ratio. Reaction condition: $P = 5.0$ MPa; $F = 25$ mL min$^{-1}$; $T = 583$ K; time on stream, 20 h. **d** Effect of total pressure. Reaction condition: $H_2/CO = 1:1$; $F = 25$ mL min$^{-1}$; $T = 583$ K; time on stream, 20 h. **e** Effect of CO partial pressure at a fixed $H_2$ pressure. Reaction conditions: $P(H_2) = 2.5$ MPa; $T = 583$ K; $F = 40$ mL min$^{-1}$; $P = 5$ MPa; time on stream, 20 h. **f** Effect of $H_2$ partial pressure at a fixed CO pressure. Reaction conditions: $P(CO) = 2.0$ MPa; $T = 583$ K; $F = 40$ mL min$^{-1}$; $P = 5$ MPa; time on stream, 20 h. C$_{2+}$ HC: C$_{2+}$ hydrocarbons; EA: ethyl acetate; AA: acetic acid; $r(CO)$: the rate of CO conversion. In all cases, the weights of $K^+$–ZnO–ZrO$_2$, H-MOR–DA–12MR and Pt–Sn/SiC catalysts were 0.66, 0.66 and 0.66 g, respectively. The experiments in each case were performed for three times. The error bar represents the relative deviation, which is within 5%.

Furthermore, the three catalysts in the tandem system were very stable, and no changes in CO conversion and ethanol selectivity were observed in 100 h of reaction (Fig. 4b).

We further investigated the temperature effect on catalytic behaviours of each catalyst component in each separated reaction. From Arrhenius plots, the apparent activation energies for methanol synthesis, methanol carbonylation and acetic acid hydrogenation were calculated to be 81, 71 and 45 kJ mol$^{-1}$ over the $K^+$–$ZnO$–$ZrO_2$, H-MOR–DA–12MR and Pt–Sn/SiC catalysts, respectively (Supplementary Fig. 13). The rate of methanol synthesis is the lowest among the three steps in a temperature range of 503–583 K. These results suggest that methanol synthesis on the $K^+$–$ZnO$–$ZrO_2$ is the rate-determining step in our tandem catalysis.

The conversion of syngas with different $H_2$/CO ratios using the $K^+$–$ZnO$–$ZrO_2$ │ H-MOR–DA–12MR │ Pt–Sn/SiC system showed that upon increasing the $H_2$/CO ratio from 0.25:1 to 1:1, the CO conversion increased from 3.2 to 5.7%, while the ethanol selectivity decreased only slightly from 75 to 70% (Fig. 4c). A further increase in the $H_2$/CO ratio further increased CO conversion, but the selectivity of ethanol decreased significantly and those of $C_2$–$C_4$ olefins, paraffins and $CO_2$ increased at the same time. Similarly, the increase in the $H_2$/CO ratio to >1:1 significantly decreased the selectivity of acetic acid and increased those of $C_2$–$C_4$ olefins and $CO_2$ over the $K^+$–$ZnO$–$ZrO_2$ │ H-MOR–DA–12MR combination (Supplementary Table 8). Thus, a higher $H_2$/CO ratio favoured CO conversion but was unbeneficial to the selectivity of ethanol or acetic acid. Therefore, the present tandem system is applicable to the transformation of syngas with a wide range of $H_2$/CO ratios, particularly suitable for that derived from biomass or coal with a $H_2$/CO ratio of 0.5:1–1:1.

The total pressure of syngas also significantly affected the catalytic performance (Fig. 4d). Hydrocarbons, in particular $C_2$–$C_4$ olefins, were the major products at a lower syngas pressure. The increase in syngas pressure from 0.5 to 2.0 MPa not only enhanced CO conversion but also increased the selectivity of ethanol from 11 to 76% at the expense of that of hydrocarbons. These results indicate that a lower pressure would retard the carbonylation of $CH_3OH$ while accelerate the MTH reaction in zeolite.

The effects of partial pressures of CO and $H_2$, denoted as $P$(CO) and $P(H_2)$, on syngas conversions with $K^+$–$ZnO$–$ZrO_2$ │ H-MOR–DA–12MR │ Pt–Sn/SiC at 583 K are displayed in Fig. 4e and f. The rate of CO conversion, $r$(CO), depended strongly on both $P$(CO) and $P(H_2)$. Further analyses reveal that the experimental data could be fitted to a Langmuir-Hinshelwood kinetic model for the reaction between adsorbed CO and dissociatively adsorbed H species (Supplementary Fig. 14 and Supplementary Note 2). This kinetic behaviour is similar to that reported for the conversion of syngas to lower olefins over a $ZnO$–$ZrO_2$/SSZ-13 catalyst, where methanol synthesis on $ZnO$–$ZrO_2$ is the rate-determining step[30]. The result here further suggests that the conversion of syngas to methanol is the rate-determining step in our tandem system. As expected, the product selectivity was also affected by $P$(CO) and $P(H_2)$. A higher $P$(CO) favoured ethanol formation, whereas a higher $P(H_2)$ was unbeneficial to ethanol selectivity, similar to the trend observed for changing the $H_2$/CO ratio.

### Interplay between different steps and compatibility

The compatibility is a crucial factor in determining the performance of the tandem catalysis. Although the three catalysts in our tandem system are separated by quartz wool, the interplay between different steps also plays a pivotal role. The carbonylation of methanol with CO involves C–C coupling and is a key step. The ratio of CO/$CH_3OH$, which is crucial to methanol carbonylation, is determined by the CO conversion in methanol synthesis, i.e., the first step. Too higher a CO conversion in the first step would lead to a lower CO/$CH_3OH$ ratio and may be detrimental to the carbonylation reaction. We found that the efficiency of methanol carbonylation on H-MOR–DA–12MR depended on the CO/$CH_3OH$ ratio and reaction temperature (Supplementary Table 9). To keep a high efficiency of methanol carbonylation, the CO/$CH_3OH$ ratio should be controlled at ≥ 300 at 503 K or ≥8.5 at 583 K. This means that the CO conversion in the first step should be controlled at a level of ~0.5% at 503 K or at 10.5% at 583 K to keep an adequate CO/$CH_3OH$ ratio for the second step. This would limit the overall CO conversion at ≤21% at 583 K by taking into account a further CO consumption in the subsequent carbonylation.

The incompatibility made Cu–Zn–Al oxide, a well-known methanol-synthesis catalyst, not a good candidate for the first step in the tandem system. At 503 K, a CO conversion of 11% was obtained with $CH_3OH$ selectivity of 95% over Cu–Zn–Al oxide alone, but the Cu–Zn–Al │ H-MOR–DA–12MR combination provided DME as the major product with very low selectivity of acetic acid (Supplementary Table 10). At 583 K, the $CH_3OH$ selectivity became low (30%) over the Cu–Zn–Al oxide and the Cu–Zn–Al │ H-MOR–DA–12MR mainly catalysed the formation of hydrocarbons (Supplementary Table 10). The control of CO conversion by decreasing the amount of Cu–Zn–Al component to keep a reasonably high CO/$CH_3OH$ ratio for the second step could offer reasonably high selectivity of acetic acid using the Cu–Zn–Al │ H-MOR–DA–12MR combination (Supplementary Table 10). The selectivities of ethanol could reach 81%, 71% and 60% at CO conversions of 1.5%, 3.8% and 5.5% at 503, 523 and 543 K, respectively, when the Pt–Sn/SiC catalyst was further combined. These performances were inferior to those with the $K^+$–$ZnO$–$ZrO_2$ │ H-MOR–DA–12MR │ Pt–Sn/SiC system.

The compatibility of the catalyst with syngas stream, which contains both CO and $H_2$, is also important for the carbonylation and hydrogenation reactions. For the normal carbonylation reaction, the catalyst works under CO atmosphere without $H_2$[21,36,41–43]. Our studies reveal that the presence of $H_2$ did not significantly change the performance of H-MOR–DA–12MR in the carbonylation of methanol (Supplementary Table 11). The catalyst underwent gradual deactivation in $CH_3OH$ carbonylation with CO in the absence of $H_2$ (Fig. 5a). The deactivation of H-MOR in carbonylation is believed to arise from coke deposition in 12-MR channels[55]. The presence of $H_2$ significantly inhibited catalyst deactivation probably by suppressing the coke deposition (Supplementary Fig. 15). No changes in methanol conversion and acetic acid selectivity were observed in 50 h of reaction (Fig. 5a). We believe that this results in the superior stability of the present tandem catalytic system. Thus, the syngas stream not only offers CO for carbonylation but also ensures high stability owing to the presence of $H_2$.

We found that the co-existence of CO affected the hydrogenation of acetic acid on supported Pt catalyst. The conversion of acetic acid and the selectivity of ethanol on the Pt/SiC were both lower than those on the Pt–Sn/SiC during the hydrogenation of acetic acid with $H_2$ alone. The use of syngas instead of $H_2$ further decreased the conversion of acetic acid and the selectivity of ethanol on the Pt/SiC catalyst (Fig. 5b). On the other hand, neither the acetic acid conversion nor the ethanol selectivity changed significantly in the presence of CO over the Pt–Sn/SiC catalyst. Thus, the presence of Sn not only improves the activity and selectivity for acetic acid hydrogenation but also eliminates the negative effect of CO. As already mentioned, the presence of Sn markedly suppresses CO chemisorption on Pt surfaces

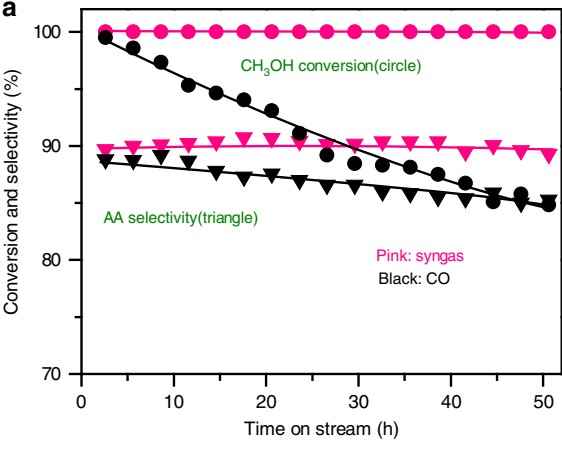

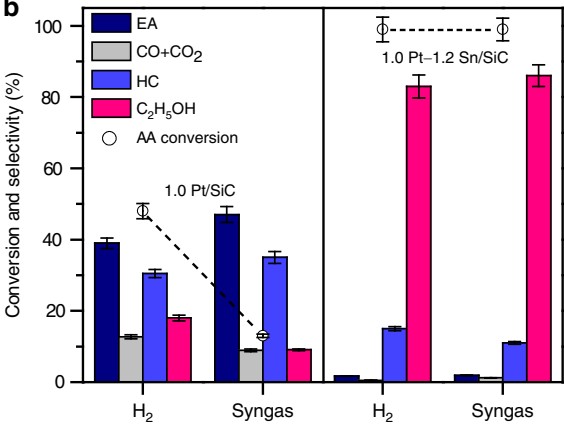

**Fig. 5 Catalyst compatibility in syngas stream. a** Catalytic performances of H-MOR-DA-12MR for methanol carbonylation in CO and syngas streams. AA: acetic acid. Reaction conditions: H-MOR-DA-12MR, 1.0 g; $P = 5.0$ MPa; $T = 583$ K; $F(CH_3OH) = 1.48$ mmol h$^{-1}$; $F(95\%CO-5\%Ar)$ or $F(48\%CO-48\%H_2-4\%Ar) = 12$ or 24 mL min$^{-1}$. **b** Catalytic performances of Pt/SiC and Pt-Sn/SiC for acetic acid hydrogenation in $H_2$ and syngas streams. EA: ethyl acetate; HC: hydrocarbons; AA: acetic acid. Reaction conditions: Pt/SiC or Pt-Sn/SiC, 0.66 g; $P = 5.0$ MPa; $T = 583$ K; $F(AA) = 1.05$ mmol h$^{-1}$; $F(H_2)$ or $F(48\%CO-48\%H_2-4\%Ar) = 12$ or 25 mL min$^{-1}$; time on stream, 20 h. The experiments in each case were performed for three times. The error bar represents the relative deviation, which is within 5%.

(Supplementary Fig. 10c), thus enabling the hydrogenation of acetic acid to proceed efficiently and selectively to ethanol.

The proximity between catalytically functional components could significantly affect catalytic behaviours in tandem catalysis[27,28,30,31,56–58]. We found that the change in the size of granules of each catalyst component in a range of 125–850 μm with the layer-by-layer (K$^+$-ZnO-ZrO$_2$|H-MOR-DA-12MR|Pt-Sn/SiC) configuration did not change the catalytic performance significantly (Supplementary Table 12). The decrease in the amount of quartz wool or the height of quartz-wool bed to separate the catalyst layers also did not change catalytic performances until the amount of quartz wool decreased to ≤ 0.015 g, which could not completely separate the catalyst layers and thus led to direct contact of catalyst components to some extent (Supplementary Table 13). The selectivity of ethanol decreased and that of lower olefins, in particular ethylene, increased as the amount of quartz wool decreased to ≤ 0.015 g. The CO$_2$ selectivity increased at the same time. This suggests that ethanol formed on Pt-Sn/SiC may undergo dehydration to

ethylene on zeolite, which has direct contact with the Pt-Sn/SiC due to the too smaller amount of quartz wool. We confirmed that ethanol could be easily converted to ethylene over H-MOR-DA-12MR under our reaction conditions (Supplementary Table 14). The formation of ethylene at the expense of ethanol has also been observed in the conversion of acetic acid upon increasing the proximity between the two components in H-MOR-DA-12MR│Pt-Sn/SiC (Supplementary Table 15). We further measured catalytic behaviours of mixed catalysts composed of three components. The use of mixed granules of three catalysts (i.e., K$^+$-ZnO-ZrO$_2$, H-MOR-DA-12MR and Pt-Sn/SiC) with sizes of 250–600 μm instead of the layer-by-layer configuration led to the disappearance of ethanol (Supplementary Table 16). C$_2$-C$_4$ olefins (mainly ethylene) became the major products; the selectivities of C$_2$-C$_4$ olefins and ethylene reached 80% and 55%, respectively. A further increase in the proximity by grinding fine powders of the three components significantly increased ethylene selectivity to 68%, although the C$_2$-C$_4$ olefin selectivity only increased slightly. The CO$_2$ selectivity also became higher by using the mixed catalysts. The excess amount of CO$_2$ possibly arises from the WGS reaction on metal oxide surfaces of the mixed catalyst. The conversion of acetic acid also led to the formation of lower olefins instead of ethanol over the mixed catalyst composed of Pt-Sn/SiC and H-MOR-DA-12MR in close contact (Supplementary Table 17). These results demonstrate that the complete separation of the three catalyst components is a key to the selective formation of ethanol.

As compared with the routes via DME and methyl acetate, by which the highest ethanol selectivity obtained is ~50%[31–35], we have achieved significantly higher ethanol selectivity (70–90%) from syngas through a single-pass route via methanol and acetic acid intermediates. Furthermore, although ethanol can be produced through three separated processes, i.e., methanol/DME synthesis, methanol/DME carbonylation and acetic acid/methyl acetate hydrogenation (Fig. 1, Routes B and C), the single-pass route reported in this work would simplify the reaction and separation/purification processes as well as the required equipment, and can thus reduce the cost and improve the operation efficiency. The tandem catalysis in one reactor can also bring about beneficial effect on the stability of zeolite catalyst for carbonylation reaction, which is known as a challenging issue[55]. For future studies, to further increase the activity by catalyst modification or reactor engineering is required. For example, to design a reactor with three catalyst layers controlled at different temperatures may overcome the CO conversion limitation. The separation of H$_2$O formed together with ethanol in the last step needs consideration to obtain anhydrous ethanol. The extractive distillation, a technique widely used for the separation of ethanol–water azeotropes derived from ethylene hydration and sugar fermentation[59], as well as membranes[60,61] may fulfil this task. An anti-corrosive reactor may also be necessary if the concentration of acetic acid intermediate becomes considerable in the large-scale process.

## Discussion

We have succeeded in designing a triple tandem catalytic system that can accomplish selective conversion of syngas into ethanol. The system is composed of potassium-modified ZnO-ZrO$_2$, zeolite H-MOR-DA-12MR and Pt-Sn/SiC in one reactor, which catalyse syngas to methanol, methanol carbonylation to acetic acid and hydrogenation of acetic acid to ethanol in tandem. The ethanol selectivity reaches 90% and 81% at CO conversions of 0.7% and 4.0% at 503 and 543 K, respectively. At a CO conversion of ~10%, the ethanol selectivity could be sustained at 64%. The system is very stable and no deactivation is observed in 100 h. We

have demonstrated that to keep the high selectivity of each step by carefully designing the corresponding catalyst is the key to obtaining high ethanol selectivity. The interplay between different steps and the compatibility of catalysts in syngas stream are also crucial. Methanol synthesis is the rate-determining step and to keep a sufficiently high $CO/CH_3OH$ ratio is important for methanol carbonylation, which determines C–C coupling. The presence of $H_2$ improves the stability of zeolite for methanol carbonylation, whereas the co-existence of CO requires careful design of catalysts for acetic acid hydrogenation to avoid the poisoning effect of CO. The present work not only presents a promising catalytic system for high-selective conversion of syngas into ethanol but also offers a method of controlling reaction selectivity by decoupling a complicated and uncontrollable reaction into well-controlled tandem reactions.

## Methods

**Synthesis of metal oxide catalysts.** The $ZnO–ZrO_2$ and $K^+–ZnO–ZrO_2$ were synthesized by a sol-gel method[30]. Typically, $Zn(NO_3)_2·6H_2O$, $Zr(NO_3)_4·5H_2O$ and citric acid were dissolved in 150 mL deionized water. $KNO_3$ was also added in the solution for the synthesis of $K^+–ZnO–ZrO_2$. The mixture was evaporated at 363 K until a viscous gel was obtained. Then, the resultant was heated to 473 K for 5 h and calcined at 773 K in air for 5 h. The typical molar ratios of Zn/Zr for the $ZnO–ZrO_2$ and K/Zn/Zr for the $K^+–ZnO–ZrO_2$ were 1:8 and 0.07:1:8, respectively, unless otherwise specified. The $ZnAl_2O_4$[31], $ZnGa_2O_4$[62] and $ZnCr_2O_4$[63] compounds were synthesized by a co-precipitation method. As an example, for the synthesis of $ZnAl_2O_4$, $Zn(NO_3)_2·6H_2O$ (2.97 g) and $Al(NO_3)_3·9H_2O$ (7.56 g) were dissolved in 100 mL deionized water. Then, an aqueous ammonia solution (25 wt%) was added dropwise into the mixed solution at room temperature until the pH reached 7. The obtained precipitate was aged for 2 h at 343 K. Then, the solid product was recovered by filtration, washing with deionized water and drying overnight at 373 K. The $ZnAl_2O_4$ compound was obtained after calcination in air at 773 K for 5 h. $Ga(NO_3)_3·6H_2O$ and $Cr(NO_3)_3·9H_2O$ were employed instead of $Al(NO_3)_3·9H_2O$ for the synthesis of $ZnGa_2O_4$[62] and $ZnCr_2O_4$[63]. $(NH_4)_2CO_3$ aqueous solution was used as the precipitant in the case of $ZnCr_2O_4$ synthesis[63]. The $ZrO_2–In_2O_3$ composite with a Zr/In molar ratio of 1:2 was prepared by a similar co-precipitation method using $Zr(NO_3)_4·5H_2O$ and $In(NO_3)_3·4.5H_2O$ as starting materials[38]. The Cu–Zn–Al oxide was synthesized by a co-precipitation method. Typically, $Cu(NO_3)_2·3H_2O$ (7.25 g), $Zn(NO_3)_2·6H_2O$ (4.46 g) and $Al(NO_3)_3·9H_2O$ (1.88 g) were dissolved in 50 mL deionized water. Then, the mixed aqueous solution and an aqueous solution of $Na_2CO_3$ (1.0 mol mL$^{-1}$) as a precipitant were simultaneously added to a glass beaker with 100 mL deionized water under vigorous stirring at 343 K. The pH value of the aqueous solution was kept at 7.0. The obtained precipitate was aged for 1 h. Then, the solid product was recovered by filtration, followed by washing thoroughly with deionized water. The product was dried overnight at 373 K, followed by calcination in air at 623 K for 5 h and reduction in $H_2$ flow at 523 K for 2 h.

**Zeolites and selective dealumination of H-MOR.** Zeolites used in this work, including SAPO-34, H-beta, H-ZSM-5, H-ZSM-35 and H-MOR, were purchased from Nankai University Catalyst Co. Selective dealumination of H-MOR with a Si/Al ratio of 13 was performed by a high-temperature steam-treatment method[31,43]. In brief, H-MOR (30–60 mesh, 3.3 g) was first pretreated by pyridine in a stream containing pyridine in $N_2$ (pyridine pressure of 2 kPa in $N_2$ gas flow with a flow rate of 70 mL min$^{-1}$) for 1 h in a quartz-tube reactor. A Py-MOR sample, whose 12-MR channel was occupied by pyridine molecules, was obtained. Then, the Py-MOR sample was treated in a 0.05 M aqueous $NaNO_3$ solution at 353 K for 2 h to allow the protons in the 8-MR channels being exchanged by $Na^+$ cations to obtain an H-MOR–Py–Na–8MR. The resulting solid was subsequently calcined at 873 K in air for 5 h to remove pyridine molecules in the 12-MR channels to obtain an H-MOR–Na–8MR sample. The steam treatment of the H-MOR–Na–8MR was conducted at 723 K for 5 h in the quartz reactor with a stream containing 26% $H_2O$ in $N_2$. The dealumination was expected to occur selectively for the Al sites in the 12-MR channels[43]. The obtained H-MOR–Na–8MR–steam sample was ion-exchanged by using $NH_4Cl$ aqueous solution (1.0 M) for three times at 353 K for 2 h. The solid $NH_4$-MOR sample was recovered by filtration and washing with deionized water, and finally calcined at 773 K in air for 5 h. The sample thus obtained was denoted as H-MOR–DA–12MR.

**Synthesis of supported metal catalysts.** Supported Pt and Pt–Sn catalysts were prepared by a chemical reduction method. SiC (β-phase, nanopowder) with a purity of 95% was purchased from Alfa Aesar. For the preparation of the Pt–Sn/SiC catalyst, SiC (1.0 g) was first dispersed in ethylene glycol (20 mL) and sonicated for 0.5 h. Aqueous solutions of $H_2PtCl_6$ (3.7 mg mL$^{-1}$) and $SnCl_2$ (4.0 mg mL$^{-1}$) were added to above solution. Then, a sodium dodecyl sulphate (SDS) solution (0.6 mg mL$^{-1}$) was added into the mixed solution and stirred for 1 h. Subsequently, $NaBH_4$

aqueous solution was added dropwise to the mixed solution and stirred at room temperature for 3 h. The solid product was recovered by filtration, washing and drying at 393 K overnight. The powdery Pt–Sn/SiC catalyst was calcined at 623 K for 4 h. The typical loadings of Pt and Sn were 1.0 wt% and 1.2 wt%, respectively. The same procedure was used for the preparation of the Pt/SiC catalyst without addition of $SnCl_2·5H_2O$. The Pd/SiC and Rh/SiC were also prepared using the same procedure using $PdCl_2$ and $RhCl_3$ aqueous solutions instead of $H_2PtCl_6$ aqueous solution. The $Cu/SiO_2$ catalyst with a Cu loading amount of 20 wt% was prepared by an impregnation method using $SiO_2$ (fused silica) and $Cu(NO_3)_2$ aqueous solution as the starting materials. After the impregnation, the sample was dried in vacuum at 323 K, followed by calcination in air at 673 K for 4 h and reduction in $H_2$ flow at 523 K for 2 h.

**Catalytic reaction.** The conversion of syngas was performed on a high-pressure fixed-bed flow reactor (titanium reactor with an inner diameter of 10 mm) designed by Xiamen Han De Engineering Co., Ltd. The catalysts, which possessed grain sizes of 250–600 μm (30–60 mesh), were used for each reaction. In the titanium reactor, the catalysts with different functions were separated by quartz wool. For example, the catalyst bed containing three layers of catalysts of $K^+–ZnO–ZrO_2$, H-MOR–DA–12MR and Pt–Sn/SiC separated by quartz wool was denoted as $K^+–ZnO–ZrO_2$｜H-MOR–DA–12MR｜Pt–Sn/SiC. Syngas containing argon with a concentration of 4%, which was used as an internal standard for the calculation of CO conversion, was introduced into the reactor. The flow rate of syngas was typically manipulated at 25 mL min$^{-1}$ (STP) unless otherwise stated. The pressure of syngas was typically 5.0 MPa. The conversion of $CH_3OH$ in CO or syngas stream was performed in the same reactor. Liquid methanol with a flow rate of 1.48 mmol h$^{-1}$ was fed into the reactor by a Series III Pump with 95%CO–5%Ar or syngas (48%$H_2$–48%CO–4%Ar) flow. The conversion of $CH_3COOH$ in $H_2$ or syngas stream was performed by feeding liquid acetic acid with a flow rate of 1.05 mmol h$^{-1}$ into the reactor by a Series III Pump with pure $H_2$ or syngas (48% $H_2$–48%CO–4%Ar) flow. The conversion of $CH_3CH_2OH$ in $N_2$ stream was performed by feeding liquid ethanol with a flow rate of 1.05 mmol h$^{-1}$ into the reactor by a Series III Pump with pure $N_2$ flow. Products were analysed by an online gas chromatograph (Ruimin GC2060, Shanghai), which was equipped with a thermal conductivity detector (TCD) and a flame ionization detector (FID). A TDX-01 packed column was connected to the TCD for the separation and analyses of Ar, CO, $CH_4$ and $CO_2$. A RT–Q–BOND–PLOT capillary column was connected to the FID for the separation and analyses of organic oxygenated compounds (including methanol, DME, ethanol, methyl acetate, acetic acid and ethyl acetate) and $C_1–C_{10}$ hydrocarbons. The product selectivity was calculated on a molar carbon basis. The selectivity of $CO_2$, which was formed by the WGS reaction, was calculated separately from CO hydrogenation products. Carbon balances were all about 95–99%. Each catalytic reaction was typically performed for three times and the relative deviation was within 5%, confirming that the catalytic performance is reproducible.

**Catalyst characterization.** X-ray diffraction (XRD) patterns were recorded on a Rigaku Ultima IV diffractometer (Rigaku, Japan). Cu Kα radiation (40 kV and 30 mA) was used as the X-ray source. X-ray fluorescence (XRF) spectroscopy was used to measure the elemental compositions of catalysts in this work. The XRF measurements were performed with a Panalytical Axois Petro XRF instrument with rhodium target (50 kV, 50 mA). Transmission electron microscopy (TEM) and line-scan energy-dispersive X-ray spectroscopy (EDS) measurements were performed on a Phillips Analytical FEI Tecnai 20 electron microscope operated at an acceleration voltage of 200 kV. The sample was dispersed ultrasonically in ethanol for 5 min, and a drop of solution was deposited onto a carbon-coated copper grid. X-ray photoelectron spectroscopy (XPS) measurements were carried out in an UHV chamber equipped with an Omicron XPS (base pressure $5 \times 10^{-10}$ torr). A monochromatized Al $K_\alpha$ X-ray source and a Sphere 2 analyser were used. After pretreated in $N_2$ at 673 K or syngas with $H_2/CO = 1$ at 583 K for 1 h in a pre-treatment chamber directly connected to the detecting chamber, the sample was transferred into the UHV detecting chamber for XPS measurements. The electron paramagnetic resonance (EPR) spectroscopy measurements were carried out on Bruker EMX-10/12 instrument at temperature of 77 K, microwave frequency of 9.43 GHz and microwave power of 19.83 mW. 1,1-Diphenyl-2-picryl-hydrazyl was used as an internal standard to quantify the intensity of EPR signals. $NH_3$ temperature-programmed desorption ($NH_3$-TPD) measurements were performed on a Micromeritics Auto Chem II 2920 instrument. Typically, the sample was pretreated in a quartz reactor with a high-purity He gas flow at 673 K for 1 h. The adsorption of $NH_3$ was performed at 373 K in an $NH_3$-He mixture (10 vol% $NH_3$) for 1 h, and TPD was performed in He flow by raising the temperature to 1173 K at a rate of 10 K min$^{-1}$. Fourier-transform infrared (FT-IR) studies were performed with a Nicolet 6700 instrument equipped with an MCT detector. The sample was pressed into a self-supported wafer and placed in an in situ IR cell. After pretreatment under vacuum at 673 K for 4 h, the sample was cooled to room temperature. Then, FT-IR spectra were recorded from 4000 to 600 cm$^{-1}$ by averaging 64 scans collected at 4 cm$^{-1}$ resolution. Pyridine-adsorbed FT-IR measurements were performed with the same instrument. After pretreatment under vacuum at 673 K for 30 min, the sample was cooled down to 423 K. Then, pyridine adsorption was carried out at 423 K for 30 min. IR spectra were collected after gaseous or weakly adsorbed pyridine molecules were removed by evacuation at 423 K. The

amounts of CO chemisorption were measured at 308 K on a Micromeritics ASAP 2020 C apparatus. $^1H$, $^{23}Na$ and $^{27}Al$ magic angle spinning nuclear magnetic resonance (MAS NMR) experiments were conducted on a Bruker Avance III 400 NMR spectrometer equipped with an 89 mm wide-bore 9.4 T superconducting magnet using 4.0 mm MAS probe tuned to $^1H$ at 400.0 MHz, $^{23}Na$ at 105.8 MHz and $^{27}Al$ at 104.3 MHz, respectively. The reported $^1H$, $^{23}Na$ and $^{27}Al$ chemical shifts were referenced to adamantane, 1.0 M NaCl and 1.0 M $Al(NO_3)_3$ aqueous solution at 1.91, 0 and 0 ppm, respectively. All the samples for NMR measurements were kept under $10^{-3}$ Pa for 3 h at 673 K, before they were cooled down to room temperature. After that, the samples were packed into 4.0 mm $ZrO_2$ rotors in a $N_2$ glove box. The description of conditions for MAS NMR: (For $^1H$) A single $\pi/2$ pulse (3.5 μs) was used for excitation; recycle delay = 1 s; spinning speed = 14.0 kHz. (For $^{23}Na$) A single pulse corresponding to a $\pi/4$ flip angle (1.7 μs) was used for excitation; recycle delay = 0.1 s; spinning speed = 14.0 kHz. (For $^{27}Al$) A single pulse corresponding to a $\pi/6$ flip angle (1.0 μs) was used for excitation; recycle delay = 0.1 s; spinning speed = 14.0 kHz. GC-MS analyses were performed using a Thermo ISQ7000 instrument with a TS-1ms capillary column. 0.05 g spent zeolite catalysts were dissolved in 0.05 mL HF solution (20%). After being neutralized with 5 wt% NaOH solution, the soluble cokes were extracted with 0.5 mL $CH_2Cl_2$ containing 20 ppm $C_2Cl_6$ (internal standard). Thermogravimetric (TG) analysis was performed in air flow on a SDT-Q600 apparatus. The sample was dried at 373 K for 2 h, then heated to 1073 K at a rate of 10 K min$^{-1}$ under air flow.

## Data availability

The data supporting the findings of this study are available from the corresponding authors on reasonable request.

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

## Acknowledgements

This work was supported by the National Key Research and Development Program of the Ministry of Science and Technology of China (No. 2017YFB0602201) and the National Natural Science Foundation of China (Nos. 91945301, 91545203, 21972116, 21433008, 21872112, 21673188 and 21690082). We acknowledge Prof. L. Peng and Y. Wen (Nanjing University, China) for performing NMR characterizations.

## Author contributions

J.K. and S.H. performed experiments for syntheses of catalysts and catalytic reactions, and analysed the experimental results. J.K. also co-wrote the paper. W.Z. performed most of the catalyst characterisations and analysed the characterisation results. Z.S. conducted the catalyst stability test. Y.L. and M.C. conducted FT-IR and XPS characterisations. Q.Z. analysed all the data and co-wrote the paper. Y.W. designed and guided the study, and co-wrote the paper. All of the authors discussed the results and reviewed the paper.

## Competing interests

The authors declare no competing interests.
