## [Peer Review File · Nature Communications]

Reviewers' comments:

Reviewer #1 (Remarks to the Author):

Kang et al. have put together a well-written communication on using triple tandem catalysts (K-ZnO-ZrO₂|H-MOR-DA-12-MR|PtSn/SiC) for direct conversion of synthesis gas into ethanol. The authors characterize each catalyst component with TEM, XRD, NH₃-TPD and XPS and use packed bed reactor studies to measure catalytic performance. Although the authors do demonstrate a few interesting findings (90% EtOH selectivity at 0.7% conversion), this reviewer is left with several key questions to better understand the complex triple tandem catalyst. A few specific comments are found below:

1. One piece of important information that is found in Table 3 of the Supplemental Information is selectivity to CO₂. The selectivity toward the byproduct CO₂ is not mentioned once in the manuscript, but is first included in Table 3 of the SI. The authors should include CO₂ selectivity in Figures 2 and 4 or at least explain why CO₂ was not included or mentioned in the main body of the manuscript. Otherwise, Figures 2 and 4 come across as misleading. Furthermore, from inspection of the tables in the SI, it appears that CO₂ selectivity increases with increasing conversion and EtOH selectivity, making it extremely important that CO₂ is included as a product within the manuscript.

2. The idea of tandem catalysts is certainly not a new one, but has found increased popularity of late because they can be used to precisely control a reaction cascade. The authors demonstrate that this catalyst system can achieve high selectivity (90%) to EtOH, but at very low conversion (0.7%). From the data included in the manuscript, it appears that EtOH selectivity decreases with increasing conversion, making it extremely difficult to scale-up or improve this triple tandem catalyst. Can the authors comment on why this tandem catalyst system is limited to high EtOH selectivity at low conversion? The authors should also comment if high EtOH selectivity can be achieved at high conversion or if this reaction scheme is thermodynamically limited and cannot be solved via catalysis.

3. The proximity between each active phase is extremely important in tandem catalysis. For a similar, two-catalyst tandem reaction, it has been shown that the grain size and proximity between the active phases can greatly influence product selectivity (Gao, P. et al. Nature Chemistry 2017, 9, 1019-1024). These findings are further supported in a detailed mathematical analysis by Weisz (Weisz, P.B., Advances in Catalysis, 1962; 13, 137-190). How did the authors control for catalyst particle size? It seems like the range of grain sizes in the reactor were rather large (250-600 μm). Were efforts made to control the amount of quartz wool used to separate the three catalysts? How reproducible are these reactor findings? If the authors run the same reaction, but vary the catalyst

grain sizes or amount of quartz wool, do the authors expect to achieve similar results as the current manuscript?

Reviewer #2 (Remarks to the Author):

Ethanol is an important clean fuel and basic chemical, which is now primarily synthesized by fermentation. The main problem of biological fermentation is low efficiency and high energy consumption for ethanol separation. The chemical synthesis of ethanol from non-petroleum resources via syngas or its derivatives such as DME and MeOH is promising. An important innovation in this manuscript is the discovery that Pt-Sn/SiC catalysts can selectively hydrogenate CH₃COOH without hydrogenating CO in the presence of syngas and acetic acid, which leads to 90% ethanol selectivity for the triple tandem catalysts. Although the findings are interesting, in my opinion, the manuscript needs major revisions and the authors should address the following points before being accepted.

1) The author's recent paper (Angew. Chem. Int. Ed. 57, 437 12012-12016 (2018)) has proposed the concept of syngas directly synthesizing ethanol over a triple tandem catalyst. The ZnAl₂O₄|H-MOR|ZnAl₂O₄ provides ethanol with a selectivity of 52%. Although the above paper has been cited in this manuscript, the authors do not directly describe the progress that has been made. I think it will cause misunderstandings to readers.

2) Although the selectivity of ethanol is high, the conversion efficiency of syngas is too low. I don't quite understand why the authors choose a low-efficiency methanol synthesis catalyst K⁺-ZnO-ZrO₂. The synthesis efficiency of conventional CuZnAlO_x methanol synthesis catalyst should be much higher. Of course, the conversion over CuZnAlO_x may not be high at 583 K due to chemical equilibrium limitations. However, this problem could be solved by separately controlling the temperature of the triple tandem catalyst. I strongly recommend that the authors compare the reactivity of CuZnAlO_x as the first layer catalyst.

3) As far as I know, some recent literatures have reported that ethanol could be synthesized by syngas over three-stage catalysts (Catal. Sci. Technol., 2019, 9, 1581) or by syngas and DME over two-stage catalysts (ChemSusChem 2010, 3, 1192-1199; J. Jpn. Petrol. Inst. 2009, 52, 357-358; Catal.Today, 2011, 164, 425-428; Catal. Sci. Technol., 2018,

8, 2087). Methyl acetate is acted as intermediate in these works. Therefore, the final products contains methanol, and the theoretical maximum selectivity of ethanol is only 66.7%. In this work, the selectivity can achieve 90%, which seems to be much higher than previous works. However, we should consider that hydrogenation of acetic acid produces an equal molar amount of water, while hydrogenation of methyl acetate is more conducive to the formation of anhydrous ethanol. In practice, methanol produced by methyl acetate can be recycled. Furthermore, the risk of corrosion

for intermediate acetic acid also exists. In my opinion, the authors should comment on these facts in the revised manuscript.

4) In the Introduction part (line 38-42), “the selectivity of ethanol is limited to 67% with one-third of carbon released as CO₂ in the biological process, because the fermentation generally proceeds via a pyruvate (C₃ sugar) intermediate, which undergoes decarboxylation to ethanol and CO₂. The chemical synthesis of ethanol from various carbon resources via syngas represents one of the most promising approaches.” In my opinion, synthesis of ethanol from syngas can also release a large amount of CO₂ in the process of syngas production.

5) In the Results part (line 131-132), “The use of Cu/SiO₂ provided higher ethanol selectivity (46%), but CH₄ was formed with a considerable selectivity (13%).” Cu/SiO₂ may be not used under the suitable reaction conditions. The temperature is too high?

6) Too many conclusions in the last part (line 240-263). This manuscript does not seem to have so many innovations. It needs to be more concise.

7) In the Method part (line 292-297), “The resulting solid was subsequently calcined at 873 K in air for 5 h to remove pyridine molecules in the 12-MR channels to obtain an H-Na-MOR sample. The steam treatment of the H-Na-MOR was conducted at 723 K for 5 h in the quartz reactor with a stream containing 26% H₂O in N₂. The dealumination occurred selectively for the Al sites in the 12-MR channels”. Despite the support of the results of FR-IR and previous works, I am very suspicious that selective dealumination of the 12-MR channels can be really achieved. On the one hand, Na⁺ will migrate at high temperatures; on the other hand, some aluminum atoms are located at the intersection of the 8-MR channels and 12-MR channels. More evidences are needed to support it.

Reviewer #3 (Remarks to the Author):

The authors have studied the conversion of syngas to ethanol using relay-type catalysis with a sandwiched fixed-bed structure. They report a maximum selectivity of 90% at otherwise low overall CO conversion < 1%. The authors also provide some characterization data of their catalysts using standard methods like XRD, NH₃-TPD, TEM etc, and end up with some catalyst compatibility studies.

I am afraid to say that this paper does not provide the necessary breakthrough warranting publication in Nature Communications. The progress made with regard to their paper in “Angewandte” (Wei Zhou et al. *Angew. Chem. Int. Ed.* 2018, 57. 12012) is marginal. Furthermore, the present paper does not provide anything conceptually new since sandwiched layers of catalysts, to some extent similar to those used here, were previously used in their work. Of course, ZnO-ZrO₂ (known to be active in methanol production) was promoted with K⁺ and a Pt/Sn catalyst was added to create a triple bed catalyst structure, but does this provide sufficient material to go for a Nature Communications paper? Increasing the ethanol selectivity from 52%, as previously reported, to 90 %

(or 81%) while encountering a decrease in the CO conversion from 6% to 0.6% (4%) just by replacing ZnAl₂O₄|H-Mor|Zn₂O₄ by ZnO-ZrO₂|H-Mor|Pt-Sn is simply not convincing. Finally, not much progress has been made with regard to a mechanistic understanding of the catalytic reactions.

I would recommend the authors provide a full story of their work (including advanced catalyst characterization and kinetic evidence) and submit to a journal devoted to catalysis.

Reviewer #4 (Remarks to the Author):

In this manuscript, Wang et al. have shown the selective conversion of syngas to ethanol through a tandem process with three catalytic components. The results present in this work can be of general interests to the catalysis community, though the CO conversion is not high. The manuscript is well structured and written. This paper can be accepted by Nature Communications after revision.

1) The characterization of PtSn/SiC show that part of the Pt species are positively charged (see Fig. S6). However, the XPS measurement was not carried out under in situ conditions. I suggest the authors to reduce the PtSn/SiC sample by H₂ and then transfer to the XPS analysis chamber to avoid contact with air. Considering the reaction conditions for syngas conversion, it is possible that Pt could be completely reduced under the reaction conditions.

2) In Figure S7, the authors show the catalytic results obtained at different temperature. It is noted that, the CO conversion almost remain unchanged when increasing the temperature from 573 to 643 K. In a recent paper (Wang et al., Sci. Adv. 2017;3: e1701290), the CO₂ conversion will increase sharply with the reaction temperature on ZnZrOx catalyst. Can the author give some explanation on the results shown in Figure S7?

3) Since the CO conversion is not very sensitive to the reaction temperature. Can the CO conversion be improved by increasing the amount of K-ZnO-ZrOx catalyst?

4) In the triple tandem catalytic system, which step is the rate-limiting step? The CO hydrogenation on K-ZnO-ZrOx or the carbonylation or the hydrogenation of acetic acid on PtSn/SiC? Some kinetic studies will help to answer this question and bring more insights on the reaction mechanism on the triple tandem system.

5) How the mass ratio of the three components influence the activity and product distributions?

6)what if the Pt loading in the Pt-sn/SiC

7) there are not details about pyridine IR study

8) the present results and catalysts (spetially for syngas to acetic gas) should be compared and discussed from those presented in patents from BP company.

Responses to Reviewers

Response to Reviewer 1

General comment: Kang et al. have put together a well-written communication on using triple tandem catalysts (K-ZnO-ZrO₂|H-MOR-DA-12-MR|PtSn/SiC) for direct conversion of synthesis gas into ethanol. The authors characterize each catalyst component with TEM, XRD, NH₃-TPD and XPS and use packed bed reactor studies to measure catalytic performance. Although the authors do demonstrate a few interesting findings (90% EtOH selectivity at 0.7% conversion), this reviewer is left with several key questions to better understand the complex triple tandem catalyst. A few specific comments are found below:

Reply: We thank the reviewer for the kind comments on our manuscript. Our replies to the comments raised by this reviewer and the corresponding revisions are described as follows.

Comment 1: One piece of important information that is found in Table 3 of the Supplemental Information is selectivity to CO₂. The selectivity toward the byproduct CO₂ is not mentioned once in the manuscript, but is first included in Table 3 of the SI. The authors should include CO₂ selectivity in Figures 2 and 4 or at least explain why CO₂ was not included or mentioned in the main body of the manuscript. Otherwise, Figures 2 and 4 come across as misleading. Furthermore, from inspection of the tables in the SI, it appears that CO₂ selectivity increases with increasing conversion and EtOH selectivity, making it extremely important that CO₂ is included as a product within the manuscript.

Reply and actions taken: We thank the reviewer for this pertinent comment. The formation of CO₂ in syngas conversion typically arises from the water-gas shift (WGS) reaction, that is, $\text{CO} + \text{H}_2\text{O} \rightarrow \text{CO}_2 + \text{H}_2$, which is initiated by water generated from CO hydrogenation. Unlike oxygenates and hydrocarbons products, which are formed by the hydrogenation of CO, CO₂ is not formed by the hydrogenation reaction but by a different type of reaction, i.e., the WGS reaction between CO and H₂O. Therefore, we did not include CO₂ selectivity in the figures in the main text of the previous version of manuscript but just listed the CO₂ selectivity in the tables in the Supplementary Information. We agree with the reviewer that it would be more straightforward to show the CO₂ selectivity in the figures in the main text. Therefore, we have added the CO₂ selectivity in Figs. 2 and 4 as well as in some tables and figures in the Supplementary Information (Supplementary Fig. 10 and Supplementary Tables 1, 3-6, 8, 10, 12-14). Brief descriptions and discussions have also been added in the main text of the revised manuscript.

Generally, the selectivity of CO₂ is determined by the catalyst used, the reaction conditions and the amount of water generated during CO hydrogenation. It is noteworthy that the formation of oxygenates as the major products during CO hydrogenation would lead to lower CO₂ selectivity because of the smaller amount of H₂O formed (most of oxygen atoms in CO are incorporated into oxygenates). On the other hand, the increase in hydrocarbon selectivity would lead to higher CO₂ selectivity because a larger amount oxygen in CO would be released as H₂O. In our case, the selectivity of CO₂ is in the range of 10-20% over the combination of K⁺-ZnO-ZrO₂ | H-MOR-DA-12MR | Pt-Sn/SiC under the conditions, where the oxygenates are the major products (Figs. 2d and 4). The increase in the selectivity of hydrocarbons at higher CO conversion increases CO₂ selectivity (Fig. 4), as the reviewer mentioned.

We have added the following sentence in the Method section: “*The selectivity of CO₂, which was formed by the WGS reaction, was calculated separately from CO hydrogenation products*” (please see Page 20, Lines 19-21; Page 21, Lines 495-496 in the created PDF file). The following sentences have been added or modified in the revised manuscript to describe the CO₂ formation in several figures and several places:

- (1) “*Over these metal oxides, CO₂ was also formed, indicating the occurrence of water-gas shift (WGS) reaction caused by H₂O formed during syngas conversions. The CO₂ selectivity, which was calculated separately from the hydrogenation products, decreased in the sequence of ZnGa₂O₄ > ZnAl₂O₄ > ZrO₂-In₂O₃ > ZnCr₂O₄ > ZnO-ZrO₂ > K⁺-ZnO-ZrO₂” (please see Page 6, Paragraph 1, the last two sentence; Page 6, Lines 116-120 in the created PDF file);*
- (2) “*The combination of K⁺-ZnO-ZrO₂|H-MOR-DA-12MR also showed the lowest selectivity of CO₂” (please see Page 7, Lines 7-8; Page 7, Lines 137-138 in the created PDF file);*
- (3) “*The change in CO₂ selectivity with metal oxide for the metal oxide|H-MOR-DA-12MR combination showed a similar trend to that for metal oxides alone” (please see Page 7, Lines 10-12; Page 7, Lines 141-143 in the created PDF file);*
- (4) “*C₂-C₄ olefins and CO₂ were the major by-products, and their selectivities were 16% and 18%, respectively” (please see Page 8, Lines 9-10; Page 8, Lines 165-166 in the created PDF file);*
- (5) “*This is because the selectivity of CH₃OH decreased significantly and those of C₂₊ hydrocarbons and CO₂ increased significantly at temperature of > 583 K over the K⁺-ZnO-ZrO₂ catalyst (Supplementary Fig. 10)” (please see Page 11, Paragraph 3, Lines 4-6; Page 12, Lines 255-257 in the created PDF file);*

(6) “A further increase in the H_2/CO ratio from 1:1 to 2:1 increased CO conversion markedly, but decreased the selectivity of ethanol and increased those of C_2 - C_4 olefins, paraffins as well as CO_2 simultaneously. Similarly, the increase in the H_2/CO ratio from 1:1 to 2:1 significantly decreased the selectivity of acetic acid and increased those of C_2 - C_4 olefins and CO_2 over the K^+ -ZnO-ZrO₂|H-MOR-DA-12MR combination (Supplementary Table 8)” (please see Page 12, Paragraph 3, Lines 4-8; Pages 12-13, Lines 276-281 in the created PDF file).

Comment 2: The idea of tandem catalysts is certainly not a new one, but has found increased popularity of late because they can be used to precisely control a reaction cascade. The authors demonstrate that this catalyst system can achieve high selectivity (90%) to EtOH, but at very low conversion (0.7%). From the data included in the manuscript, it appears that EtOH selectivity decreases with increasing conversion, making it extremely difficult to scale-up or improve this triple tandem catalyst. Can the authors comment on why this tandem catalyst system is limited to high EtOH selectivity at low conversion? The authors should also comment if high EtOH selectivity can be achieved at high conversion or if this reaction scheme is thermodynamically limited and cannot be solved via catalysis.

Reply and actions taken: We thank the reviewer for these important questions and comments. To answer the questions by the reviewer, we have performed thermodynamic analyses for the related reactions. Our tandem catalytic system is mainly composed of syngas to methanol, methanol carbonylation to acetic acid and hydrogenation of acetic acid to ethanol. The three catalyst components are separated with each other in one reactor, and thus the thermodynamics is different from that for the direct conversion of syngas to ethanol. The results of thermodynamic analyses for these reactions are shown in Supplementary Fig. 1. All the three reactions as well as the direct conversion of syngas into ethanol are thermodynamically more feasible at lower temperatures. A higher temperature is particularly unfavourable to the conversion of syngas to methanol. However, the methanol carbonylation and acetic acid hydrogenation usually need to proceed at 500-600 K because of the kinetic requirement. Thus, the whole process would be limited by the methanol synthesis, i.e., the first step. Our calculations show that the CO equilibrium conversions for the tandem catalysis for ethanol synthesis are 38% and 12% at 550 and 600 K, respectively, under syngas ($H_2/CO = 1$) pressure of 5.0 MPa (Supplementary Fig. 1). Although the CO conversion in our present tandem catalysis is thermodynamically limited at higher temperatures, ethanol selectivity can be high by the controlled tandem catalysis. For example, over our optimized system, i.e.,

K^+ -ZnO-ZrO₂|H-MOR-DA-12MR|Pt-Sn/SiC, CO conversion and ethanol selectivity could reach 9.7% and 64%, respectively, at 583 K under syngas ($H_2/CO = 1$) pressure of 5.0 MPa. This CO conversion is still lower than the equilibrium conversion predicted by thermodynamic calculations (CO equilibrium conversion to ethanol via tandem catalysis is 19% at 583 K, and thus can be further enhanced by catalyst development. Moreover, we believe that there is a large room to further increase the CO conversion while keeping high ethanol selectivity by developing efficient catalysts that can work at relatively lower temperatures to overcome the thermodynamic limitation. For example, the CO equilibrium conversion can reach 38% at 550 K.

It is noteworthy that besides the thermodynamics, we have found that the ratio of CO to methanol (CO/CH₃OH) should be controlled to ensure a high efficiency of methanol carbonylation reaction, because CO is a key reactant in this reaction. This means that the CO conversion in the first step, i.e., hydrogenation of CO to methanol, should be controlled. For example, under a syngas ($H_2/CO = 1$) pressure of 5 MPa and reaction temperature of 583 K, the CO/CH₃OH ratio should be ≥ 8.5 to obtain a high selectivity of acetic acid over the H-MOR-DA-12MR catalyst (Supplementary Table 9). Thus, the CO conversion in the first step should be controlled at $\leq 10.5\%$ at 583 K. Otherwise, the CO/CH₃OH ratio over the carbonylation catalyst would be lower and C₂₊ hydrocarbons would be formed instead of acetic acid. This would limit the overall CO conversion of syngas to ethanol $\leq 21\%$, by taking into account a further CO consumption in the subsequent carbonylation of methanol. The selectivity of methyl acetate would also become higher at the expense of acetic acid when the CO/CH₃OH ratio becomes higher, in particular at a lower temperature (Supplementary Table 9).

In short, although the CO conversion for our tandem catalysis route is limited by thermodynamics and by the requirement of keeping a sufficiently high CO/CH₃OH ratio for methanol carbonylation, this route offers an opportunity to control the product selectivity, and thus is promising for obtaining high ethanol selectivity. Moreover, the CO conversion obtained with the present system is still lower than the theoretical maximum predicted. The development of more efficient catalysts can further improve the performance for the present tandem catalysis strategy.

Based on these results and discussion described above, we have added the following figures, tables and the corresponding descriptions in the revised manuscript:

- (1) The thermodynamic analyses have been added in Supplementary Fig. 1. The following sentences have been added to briefly discuss the thermodynamic analyses in the revised manuscript: “*Thermodynamic analyses reveal that all the three tandem steps, i.e., methanol synthesis, methanol carbonylation and acetic acid hydrogenation, are thermodynamically more feasible at lower temperatures*”

(Supplementary Fig. 1). A higher temperature is particularly unfavourable to the conversion of syngas to methanol. However, methanol carbonylation and acetic acid hydrogenation usually need to proceed at 500-600 K because of the kinetic requirement^{35,36}. The CO equilibrium conversions for the tandem catalysis for ethanol synthesis were evaluated to be 38% and 12% at 550 and 600 K, respectively, under a syngas ($H_2/CO = 1$) pressure of 5.0 MPa (Supplementary Fig. 1)” (please see from Page 5-the last paragraph to Page 6-Line 3; Pages 5-6, Lines 100-108 in the created PDF file).

- (2) We have also added the discussion on the relationship between CO conversion in the first step and the efficiency of methanol carbonylation in the second step as follow: “The compatibility is a crucial factor in determining the performance of the tandem catalysis. Although the three catalysts in our tandem system are separated by quartz wool, the interplay between different steps also plays a pivotal role. The carbonylation of methanol with CO involves C-C coupling and is a key step. The ratio of CO/CH_3OH , which is crucial to methanol carbonylation, is determined by the CO conversion in methanol synthesis, i.e., the first step. Too higher a CO conversion in the first step would lead to a lower CO/CH_3OH ratio and may be detrimental to the carbonylation reaction. We found that the efficiency of methanol carbonylation on H-MOR-DA-12MR depended on the CO/CH_3OH ratio and reaction temperature (Supplementary Table 9). The CO/CH_3OH ratio should be controlled at ≥ 300 at 503 K, while the ratio could be ≥ 8.5 at 583 K to keep a high efficiency of methanol carbonylation. This means that the CO conversion in the first step should be controlled at a level of $\sim 0.5\%$ at 503 K, while it could be 10.5% at 583 K to keep an adequate CO/CH_3OH ratio for the second step. This would limit the overall CO conversion at $\leq 21\%$ at 583 K by taking into account a further CO consumption in the subsequent carbonylation” (please see from Pages 13-the last paragraph to Page 14-Paragraph 1; Pages 14, Lines 306-320 in the created PDF file).
- (3) The following sentence has been added in the Conclusion section: “the keeping of a sufficiently high CO/CH_3OH ratio plays a pivotal role in methanol-carbonylation step, which determines the C-C coupling” (please see from Page 17, Paragraph 2, Lines 11-13; Page 18, Lines 409-410 in the created PDF file).

Comment 3: The proximity between each active phase is extremely important in tandem catalysis. For a similar, two-catalyst tandem reaction, it has been shown that the grain size and proximity between the active phases can greatly influence product selectivity (Gao, P. et al. Nature Chemistry 2017, 9, 1019-1024). These findings are

further supported in a detailed mathematical analysis by Weisz (Weisz, P.B., *Advances in Catalysis*, 1962, 13, 137-190). How did the authors control for catalyst particle size? It seems like the range of grain sizes in the reactor were rather large (250-600 μm). Were efforts made to control the amount of quartz wool used to separate the three catalysts? How reproducible are these reactor findings? If the authors run the same reaction, but vary the catalyst grain sizes or amount of quartz wool, do the authors expect to achieve similar results as the current manuscript?

Reply and actions taken: As mentioned by the reviewer, the proximity between different functional components is an important factor in tandem or bifunctional catalysis, and this point has been confirmed in several published papers including our previous papers and the references mentioned by the reviewer (Refs. 26, 27, 29, 30 already mentioned in the manuscript; Ref. 53: Weisz, P. B., *Adv. Catal.* **13**, 137-190 (1962); Ref. 54: J. Zečević, et al., *Nature* **528**, 245-248 (2015); Ref. 55: Gao, P. et al. *Nat. Chem.* **9**, 1019-1024 (2017). The later three references have been newly added in the revised manuscript). In the present work, we have packed three catalyst components with different functions layer-by-layer to perform the tandem catalytic transformation from syngas to ethanol via methanol and acetic acid intermediates. This means that, unlike the situation in the references, where the catalyst components are mixed with each other in most cases, the three catalyst components in our work are separated by quartz wool and they work relatively independently. To answer the questions and comments raised by the reviewer, we have performed the following new experiments.

First, we have investigated the effect of sizes of catalyst granules on catalytic behaviours. Typically, we have used catalyst granules with sizes of 250-600 μm (30-60 mesh) and loaded the catalyst components layer-by-layer in a reactor with the inner diameter of 10 mm. This is commonly used for fix-bed catalytic reactions, which can be free of mass-transfer limitation. We have compared the catalytic performances of catalysts with the same layer-by-layer configuration in the reactor but varied grain sizes of 600-850 μm (20-30 mesh), 250-600 μm (30-60 mesh), 180-250 μm (60-80 mesh) and 125-180 μm (80-120 mesh) for each component. The result has been added in Supplementary Table 12. No significant differences in both CO conversions and product selectivities have been observed by changing the sizes of catalyst granules. Therefore, the size of granules for each catalyst component does not significantly affect the catalytic behaviours of our tandem catalytic system probably because the three catalyst components are separated with each other.

Second, we have changed the amount of quartz wool used to separate the catalyst components and investigated its effect on catalytic behaviours. The result has been added in Supplementary Table 13. When the amount of quartz wool was ≥ 0.015 g,

corresponding to a height of 1.6 mm, both CO conversions and product selectivities did not change significantly upon changing the height of quartz wool layer in the reactor. On the other hand, when the amount of quartz wool was <0.015 g, which could not completely separate different catalyst components and caused their direct contact to some extent, the selectivity of ethanol became lower and that of C₂-C₄ olefins increased. The selectivity of CO₂ also became higher at the same time (Supplementary Table 13). This suggests that the complete separation of catalyst components is necessary for obtaining high ethanol selectivity and for avoiding the possibly increased WGS reaction.

Third, we have measured catalytic behaviours of mixed catalysts composed of three components instead of placing these components layer-by-layer in the reactor. The proximity between the components is closer in this case. Ethanol disappeared when either the mixed granules or the mixed fine powders were loaded into the reactor. At the same time, ethylene was formed as a major product with a selectivity of 55-68% and the selectivity of C₂-C₄ olefins reached ~80% (Supplementary Table 14). The selectivity of CO₂ also increased simultaneously. We speculate that ethylene probably arises from the dehydration of ethanol over the zeolite that is in contact with other components. This result further suggests the importance of separation of the three catalyst components in ethanol formation.

Based on the above new results and discussion, we have added the following new paragraph to discuss the effects of granule size and proximity in the revised manuscript: *“The proximity between catalytically functional components could significantly affect catalytic behaviours in tandem catalysis^{26,27,29,30,53-55}. We found that the change in the size of granules of each catalyst component in a range of 125-850 μm with the layer-by-layer (K⁺-ZnO-ZrO₂|H-MOR-DA-12MR|Pt-Sn/SiC) configuration did not change the catalytic performance significantly (Supplementary Table 12). The decrease in the amount of quartz wool or the height of the quartz-wool bed to separate the catalyst layers also did not change catalytic performances until the amount of quartz wool decreased to ≤ 0.015 g, which could not completely separate the catalyst layers and thus led to direct contact of catalyst components to some extent (Supplementary Table 13). The selectivity of ethanol decreased and that of lower olefins, in particular ethylene, increased as the amount of quartz wool decreased to ≤ 0.015 g. The selectivity of CO₂ increased at the same time. We further measured catalytic behaviours of mixed catalysts composed of three components. The use of mixed granules of three catalyst components (i.e., K⁺-ZnO-ZrO₂, H-MOR-DA-12MR and Pt-Sn/SiC) with sizes of 250-600 μm instead of the layer-by-layer configuration led to the disappearance of ethanol in the product (Supplementary Table 14). C₂-C₄ olefins (mainly ethylene) became the major products;*

the selectivities of C₂-C₄ olefins and ethylene reached 80% and 55%, respectively. A further increase in the proximity by grinding the fine powders of the three components significantly increased the ethylene selectivity to 68%, although the C₂-C₄ olefin selectivity only increased slightly. The CO₂ selectivity also became higher by using the mixed catalysts. We speculate that ethylene is formed by the dehydration of ethanol on acidic zeolite surfaces of the mixed catalyst. The excess amount of CO₂ possibly arises from the WGS reaction on metal oxide surfaces of the mixed catalyst. These results indicate that the complete separation of the three catalyst components is a key to the selective formation of ethanol” (please see from Page 15-Paragraph 3 to Page 16-Paragraph 1; Pages 16-17, Lines 360-383 in the created PDF file).

Regarding the reproducibility, our experimental results are reproducible and the relative deviation is less than 5% in all cases. We have added this information in the section of Methods in the revised manuscript: “*The catalytic reactions were typically performed three times and the relative deviation was within 5% in each case*” (please see Page 20, Paragraph 1, the last sentence; Page 21, Lines 496-497 in the created PDF file).

Response to Reviewer 2

General comment: Ethanol is an important clean fuel and basic chemical, which is now primarily synthesized by fermentation. The main problem of biological fermentation is low efficiency and high energy consumption for ethanol separation. The chemical synthesis of ethanol from non-petroleum resources via syngas or its derivatives such as DME and MeOH is promising. An important innovation in this manuscript is the discovery that Pt-Sn/SiC catalysts can selectively hydrogenate CH₃COOH without hydrogenating CO in the presence of syngas and acetic acid, which leads to 90% ethanol selectivity for the triple tandem catalysts. Although the findings are interesting, in my opinion, the manuscript needs major revisions and the authors should address the following points before being accepted.

Reply: We appreciate the constructive comments raised by this reviewer to improve the quality of our manuscript. We have largely revised our manuscript according to the comments raised by this reviewer. We believed that many key points have now been addressed with more evidence in the revised manuscript. Our replies to the comments raised by this reviewer and the corresponding revisions are described as follows.

Comment 1: The author's recent paper (Angew. Chem. Int. Ed. 57, 437 12012-12016 (2018)) has proposed the concept of syngas directly synthesizing ethanol over a triple tandem catalyst. The ZnAl₂O₄|H-MOR|ZnAl₂O₄ provides ethanol with a selectivity

of 52%. Although the above paper has been cited in this manuscript, the authors do not directly describe the progress that has been made. I think it will cause misunderstandings to readers.

Reply and actions taken: As mentioned by the reviewer, we have published a paper on the conversion of syngas to C₂ oxygenates using relay-catalysis concept (W. Zhou, et al., *Angew. Chem. Int. Ed.* **57**, 12012-12016 (2018)). This paper had been cited as Ref. 29 in the previous version of manuscript (Ref. 30 in the revised manuscript). In that paper, we mainly focused on the synthesis of methyl acetate from syngas. Dimethyl ether (DME) was the key reaction intermediate and the selectivity of methyl acetate reached ~85%. Ethanol is a more versatile chemical having many applications in chemical and energy industries. Although we also extended the method to ethanol formation using a three-component combination of ZnAl₂O₄|H-MOR|ZnAl₂O₄, but the synthesis of ethanol was not very successful and its selectivity was only ~50%. Actually, the theoretical maximum selectivity of ethanol is 67% by the relay catalysis via DME intermediate as reported in our previous paper. The present work has succeeded in developing a new tandem catalytic system for selective synthesis of ethanol from syngas via methanol and acetic acid intermediates. The selectivity of ethanol reaches 90-81%, which can hardly be achieved over conventional catalysts because of the difficulty in controlling C-C coupling. Our present manuscript offers an effective method, i.e., the reaction (CH₃OH carbonylation)-controlled method, to control the C-C coupling to form ethanol. Significant insights have been reported to guide future design of tandem catalytic systems for C1 chemistry. For examples, we demonstrate that the interplay between methanol synthesis and methanol carbonylation plays a pivotal role in controlling the CO/CH₃OH ratio in the second catalyst bed and thus determining the efficiency of carbonylation. The presence of H₂ stabilises the zeolite catalyst for carbonylation, whereas the catalyst for hydrogenation of acetic acid should be carefully designed to be compatible in syngas atmosphere. Not only the target product and catalyst components but also the key intermediates and reaction paths in the present manuscript are different from those reported in the previous paper. Therefore, we believe that the present work represents a new advance in C1 chemistry.

We agree with the reviewer that the description of the difference of the present work from the previous paper in more detail is helpful to readers to understand the progress of the present work. The following sentences have been added in the revised manuscript: *“A few recent studies have adopted this methodology for the synthesis of C₂ oxygenates. Very recently, we succeeded in synthesizing methyl acetate from syngas by tandem catalysis via DME intermediate³⁰. Although ethanol could be obtained by using a three-component catalytic system composed of layer-by-layer*

ZnAl₂O₄, H-MOR and ZnAl₂O₄, the selectivity of ethanol was only ~50%” (please see from Page 4-the last line to Page 5-Line 4; Page 5, Lines 79-83 in the created PDF file). Furthermore, the following sentence has been added: “Here, we report the design of new tandem catalytic systems with high ethanol selectivity” (please see Page 5, Paragraph 2, Line 1; Page 5, Line 88 in the created PDF file).

Comment 2: Although the selectivity of ethanol is high, the conversion efficiency of syngas is too low. I don't quite understand why the authors choose a low-efficiency methanol synthesis catalyst K⁺-ZnO-ZrO₂. The synthesis efficiency of conventional CuZnAlOx methanol synthesis catalyst should be much higher. Of course, the conversion over CuZnAlOx may not be high at 583 K due to chemical equilibrium limitations. However, this problem could be solved by separately controlling the temperature of the triple tandem catalyst. I strongly recommend that the authors compare the reactivity of CuZnAlOx as the first layer catalyst.

Reply and actions taken: We thank the reviewer for this pertinent comment. Our tandem catalysis for the synthesis of ethanol from syngas is composed of three steps including methanol synthesis, methanol carbonylation and acetic acid hydrogenation. Besides the optimization of each step, the interplay between the three steps also needs consideration, although the three catalyst components are separated by quartz wool. In particular, the carbonylation of methanol with CO involves C-C coupling and the successful C-C coupling through the reaction of methanol with CO to acetic acid is a key step to obtaining high ethanol selectivity. The ratio of CO/CH₃OH, which is important to the carbonylation, is determined by the CO conversion in the methanol synthesis, i.e., the first step. Too higher a CO conversion in the first step would lead to a lower CO/CH₃OH ratio and may be detrimental to the carbonylation reaction. We adopted zeolite H-MOR (H-MOR-DA-12MR) for methanol carbonylation and found that the efficiency of methanol carbonylation really depended on the CO/CH₃OH ratio and also reaction temperature (Supplementary Table 9). For examples, the CO/CH₃OH ratio should be controlled at ≥ 300 at 503 K, while the ratio could be ≥ 8.5 at 583 K over H-MOR-DA-12MR. This means that at H₂/CO ratio of 1, CO conversion in the first step should be controlled at a level of ~0.33% to keep an adequate CO/CH₃OH ratio for the second step, while the CO conversion in the first step could be 10.5% at 583 K to enable the high efficiency of methanol carbonylation. By considering the balance between the CO conversion and the efficiency of carbonylation of methanol, we have chosen K⁺-ZnO-ZrO₂ which is not an excellent methanol-synthesis catalyst from the viewpoint of activity but can work at relatively higher temperatures with high CH₃OH selectivity, as our first-step catalyst.

Nevertheless, we have followed the suggestion by the reviewer and used the Cu-Zn-Al oxide as the first-step catalyst. Cu-Zn-Al oxide is a well-known methanol-synthesis catalyst that can work under mild temperatures. At 503 K, a CO conversion of 11% was obtained with an excellent CH₃OH selectivity of 95%. However, the combination of Cu-Zn-Al with H-MOR-DA-12MR (Cu-Zn-Al|H-MOR-DA-12MR) only resulted in a very low selectivity of acetic acid and DME was formed as the major product with a selectivity of 60% (Supplementary Table 10). This confirms that the carbonylation of methanol is inhibited under such conditions. On the other hand, at 583 K, the CH₃OH selectivity was low (30%) over the Cu-Zn-Al catalyst, although CO conversion was high (33%). The Cu-Zn-Al|H-MOR-DA-12MR combination also led to a very low selectivity of acetic acid (Supplementary Table 10). Therefore, although higher CO conversion and CH₃OH selectivity could be obtained in the first step at lower temperatures using the Cu-Zn-Al catalyst, this catalyst is not a good candidate for the tandem catalysis because of the unmatched reaction conditions with the methanol carbonylation. On the other hand, the K⁺-modified ZnO-ZrO₂ is a highly selective catalyst for methanol synthesis in a wide temperature range and the CH₃OH selectivity of is 93-99% at 503-583 K. Thus, the K⁺-ZnO-ZrO₂|H-MOR-DA-12MR|Pt-Sn/SiC combination has shown much better ethanol selectivity, although CO conversion is limited. By increasing the amount of K⁺-ZnO-ZrO₂ in the tandem catalytic system, we could obtain a CO conversion of 9.7% at ethanol selectivity of 64% at 583 K (Supplementary Table 6).

We have further attempted to control the CO conversion by decreasing the amount of Cu-Zn-Al component to keep sufficiently high CO/CH₃OH ratios for the second-step reaction at several selected temperatures. We could obtain reasonably high selectivity of acetic acid using the Cu-Zn-Al|H-MOR-DA-12MR combination and high selectivity of ethanol with the Cu-Zn-Al|H-MOR-DA-12MR|Pt-Sn/SiC. The ethanol selectivity reached 81%, 71% and 60% at CO conversions of 1.5%, 3.8% and 5.5% at 503, 523 and 543 K, respectively (Supplementary Table 10). These performances are still inferior to those obtained with the Cu-Zn-Al|H-MOR-DA-12MR|Pt-Sn/SiC. All these results suggest the importance of compatibility of different catalyst components in tandem catalysis for the conversion of syngas to ethanol.

Based on the new experimental results and discussion, we have added the following two paragraphs in the revised manuscript to describe the compatibility: *“The compatibility is a crucial factor in determining the performance of the tandem catalysis. Although the three catalysts in our tandem system are separated by quartz wool, the interplay between different steps also plays a pivotal role. The*

carbonylation of methanol with CO involves C-C coupling and is a key step. The ratio of CO/CH₃OH, which is crucial to methanol carbonylation, is determined by the CO conversion in methanol synthesis, i.e., the first step. Too higher a CO conversion in the first step would lead to a lower CO/CH₃OH ratio and may be detrimental to the carbonylation reaction. We found that the efficiency of methanol carbonylation on H-MOR-DA-12MR depended on the CO/CH₃OH ratio and reaction temperature (Supplementary Table 9). The CO/CH₃OH ratio should be controlled at ≥ 300 at 503 K, while the ratio could be ≥ 8.5 at 583 K to keep a high efficiency of methanol carbonylation. This means that the CO conversion in the first step should be controlled at a level of $\sim 0.5\%$ at 503 K, while it could be 10.5% at 583 K to keep an adequate CO/CH₃OH ratio for the second step. This would limit the overall CO conversion at $\leq 21\%$ at 583 K by taking into account a further CO consumption in the subsequent carbonylation” (please see from Page 13-Paragraph 3 to Page 14-Paragraph 1; Page 14, Lines 306-320 in the created PDF file); “The incompatibility made Cu-Zn-Al oxide, a well-known methanol-synthesis catalyst, not a good candidate for the first step in our tandem catalytic system. At 503 K, a CO conversion of 11% was obtained with CH₃OH selectivity of 95% over Cu-Zn-Al oxide alone, but the Cu-Zn-Al|H-MOR-DA-12MR combination only provided DME as the major product with very low selectivity of acetic acid (Supplementary Table 10). At 583 K, the CH₃OH selectivity was low (30%) over the Cu-Zn-Al catalyst, although CO conversion was quite high (33%). The Cu-Zn-Al|H-MOR-DA-12MR combination mainly catalysed the formation of hydrocarbons at 583 K (Supplementary Table 10). The control of CO conversion by decreasing the amount of Cu-Zn-Al component to keep a reasonably high CO/CH₃OH ratio for the second step could offer reasonably high selectivity of acetic acid using the Cu-Zn-Al|H-MOR-DA-12MR combination (Supplementary Table 10). Ethanol selectivities could reach 81%, 71% and 60% at CO conversions of 1.5%, 3.8% and 5.5% at 503, 523 and 543 K, respectively, when the Pt-Sn/SiC catalyst was further combined. These performances were inferior to those obtained with the K⁺-ZnO-ZrO₂|H-MOR-DA-12MR|Pt-Sn/SiC system” (please see Page 14-Paragraph 2; Pages 14-15, Lines 321-335 in the created PDF file). The method for the preparation of Cu-Zn-Al oxide has also been added in the Methods section (please see Page 18, Paragraph 2, Lines 8-1 from bottom; Page 19, Lines 434-441 in the created PDF file).

We agree with the reviewer that to control the three catalyst layers at different temperatures may be promising for overcoming the CO conversion limitation. However, this needs reactor engineering and will be the task of future studies. Therefore, we have added the following sentences in the revised manuscript to briefly mention this point: “For future studies, to further increase the activity by catalyst

modification or reactor engineering is required. For example, to design a reactor with three catalyst layers controlled at different temperatures may overcome the CO conversion limitation” (please see from Page 16-Line 2-from bottom to Page 17-Line 2; Page 17, Lines 387-390 in the created PDF file).

Comment 3: As far as I know, some recent literatures have reported that ethanol could be synthesized by syngas over three-stage catalysts (Catal. Sci. Technol., 2019, 9, 1581) or by syngas and DME over two-stage catalysts (ChemSusChem 2010, 3, 1192-1199; J. Jpn. Petrol. Inst. 2009, 52, 357-358; Catal. Today, 2011, 164, 425-428; Catal. Sci. Technol., 2018, 8, 2087). Methyl acetate is acted as intermediate in these works. Therefore, the final products contains methanol, and the theoretical maximum selectivity of ethanol is only 66.7%. In this work, the selectivity can achieve 90%, which seems to be much higher than previous works. However, we should consider that hydrogenation of acetic acid produces an equal molar amount of water, while hydrogenation of methyl acetate is more conducive to the formation of anhydrous ethanol. In practice, methanol produced by methyl acetate can be recycled. Furthermore, the risk of corrosion for intermediate acetic acid also exists. In my opinion, the authors should comment on these facts in the revised manuscript.

Reply and actions taken: We thank the reviewer for pointing out some related references. A recent paper reported that syngas could be converted to ethanol by using a combination of CuZnO_x, CuH-MOR and CuZnO_x with a sandwich configuration, but the ethanol selectivity was only ~15% at a CO conversion of ~8% (Catal. Sci. Technol. 9, 1581 (2019), cited as Ref. 31 in the revised manuscript). The other four papers from Prof. Tsubaki’s group reported ethanol synthesis from DME and syngas via a two-stage reaction composed of DME carbonylation on zeolite (HMOR, CuH-MOR or FER) and hydrogenation of methyl acetate on a Cu/ZnO or CuZnAl catalyst. The selectivity of ethanol is typically lower than 50%. Since these papers reported similar work, we have cited three of them as Refs. 32-34 in the revised manuscript. Correspondingly, we have added the following sentences in the Introduction to briefly introduce the results in these references: “A sandwich-configuration catalytic system composed of CuZnO_x, CuH-MOR, CuZnO_x was also reported for the conversion of syngas, but the ethanol selectivity was only 15%³¹. The co-feeding of DME and syngas over dual-bed catalysts for DME carbonylation and methyl acetate hydrogenation could provide ethanol with selectivity of <50%³²⁻³⁴” (please see Page 5, Lines 4-8; Page 5, Lines 83-87 in the created PDF file).

As pointed out by the reviewer, ethanol can be synthesized by a multi-process route via DME synthesis from syngas, DME carbonylation to methyl acetate and

methyl acetate hydrogenation. Methanol is formed together with ethanol and the maximum single-pass ethanol selectivity is 67%. This multi-process route has attracted much attention in both fundamental and practical studies. However, we think that it is of high significance to develop a single-pass conversion of syngas to ethanol with higher selectivity. This is the aim of the present work.

Regarding the problem of H₂O formation in our route, it is noteworthy that H₂O is always formed as a by-product if syngas is used as the starting reactant and ethanol is the target product, although some routes may not involve the direct separation of H₂O from ethanol. The separation of H₂O from ethanol should be considered for future commercialization of our route to pursue high single-pass ethanol selectivity. For example, the extractive distillation is an effective method developed to obtain pure ethanol from ethanol-water azeotrope, which is usually produced in the processes of ethylene hydration or sugar fermentation in industry (García-Herreros, P., Gómez J. M. *Ind. Eng. Chem. Res.* **50**, 3977-3985 (2019), Ref. 56 in the revised manuscript). In addition, some novel membranes, such as the membrane incorporated with MOF filler with high water adsorption capacity (Li, Q. et al. *J. Membrane Sci.* **544**, 68-78 (2017), Ref. 57 in the revised manuscript) and the membrane of microporous ionized graphene oxide (Fang, C. et al. *Carbon* **136**, 262-269 (2018), Ref. 58 in the revised manuscript), have also been developed in recent years for the separation of H₂O from ethanol with high efficiency.

Concerning the possible corrosion problem caused by acetic acid, we think that acetic acid is present as an intermediate and its concentration should be limited in our system. Therefore, its damage to the reactor is limited as compared to that in the direct synthesis of acetic acid as the target product. Anti-corrosive reactors made of zirconium or Hastelloy, which have been used in the industrial production of acetic acid may be considered for large-scale process in the future if the concentration of acetic acid becomes too high.

We have added the following new paragraph in the revised manuscript to discuss the issues mentioned above: “As compared to the routes (including both direct^{30,31} and indirect routes³²⁻³⁴) via DME and methyl acetate, by which the highest ethanol selectivity obtained is ~50%, we have achieved significantly higher ethanol selectivity (70-90%) from syngas through a single-pass route via methanol and acetic acid intermediates. For future studies, to further increase the activity by catalyst modification or reactor engineering is required. For example, to design a reactor with three catalyst layers controlled at different temperatures may overcome the CO conversion limitation. The separation of H₂O formed together with ethanol in the last step needs consideration to obtain anhydrous ethanol. The extractive distillation, a technique widely used for the separation of ethanol-water azeotropes derived from

*ethylene hydration and sugar fermentation*⁵⁶, as well as membranes^{57,58} may fulfil this task. An anti-corrosive reactor may also be necessary if the concentration of acetic acid intermediate becomes considerable in the large-scale process” (**please see from Page 16-Paragraph 2 to Page 17-Paragraph 1; Page 17, Lines 384-395 in the created PDF file**).

Comment 4: In the Introduction part (line 38-42), “the selectivity of ethanol is limited to 67% with one-third of carbon released as CO₂ in the biological process, because the fermentation generally proceeds via a pyruvate (C3 sugar) intermediate, which undergoes decarboxylation to ethanol and CO₂. The chemical synthesis of ethanol from various carbon resources via syngas represents one of the most promising approaches.” In my opinion, synthesis of ethanol from syngas can also release a large amount of CO₂ in the process of syngas production.

Reply and actions taken: We agree with the reviewer that CO₂ may be released during the production of syngas, in particular from coal. Because this is not the main point of this work, we have removed the following sentence in the previous version of manuscript to shorten the Introduction: “Moreover, the selectivity of ethanol is limited to 67% with one-third of carbon released as CO₂ in the biological process, because the fermentation generally proceeds via a pyruvate (C3 sugar) intermediate, which undergoes decarboxylation to ethanol and CO₂.⁵”. We just mentioned the limited ethanol selectivity and efficiency. In the revised manuscript, we have modified the following sentence: “The current production of ethanol relies on the fermentation of sugars, but this process suffers from the competition with food supply and the high energy-consumption in product separation and purification” into “*The current production of ethanol relies on the fermentation of sugars, but this process suffers from the competition with food supply, the high energy-consumption in product separation and purification, and limited efficiency and ethanol selectivity*⁶” (**please see Page 3, Lines 7-10; Page 3, Lines 35-38 in the created PDF file**).

Comment 5: In the Results part (line 131-132), “The use of Cu/SiO₂ provided higher ethanol selectivity (46%), but CH₄ was formed with a considerable selectivity (13%).” Cu/SiO₂ may be not used under the suitable reaction conditions. The temperature is too high?

Reply and actions taken: We have also measured the tandem catalytic system with Cu/SiO₂ as the catalyst for hydrogenation of acetic acid at lower temperatures. The result has been displayed in Supplementary Table 3. At 583 K, the typically temperature used in this work, CO conversion and ethanol selectivity were 7.8% and 46%, respectively, significantly lower than those for the system with Pt-Sn/SiC as the

hydrogenation catalyst (CO conversion 5.7%, ethanol selectivity 70%). When the temperature decreased to 503 and 543 K, not only the CO conversion but the ethanol selectivity also decreased (Supplementary Table 3). For example, at 543 K the CO conversion and ethanol selectivity were 4.2% and 34%, respectively, whereas the CO conversion and ethanol selectivity were 4.0% and 81%, respectively, at this temperature using Pt-Sn/SiC as the hydrogenation catalyst. A larger fraction of acetic acid could not be hydrogenated to ethanol by using Cu/SiO₂ at a lower temperature. Therefore, it becomes clear that the Cu/SiO₂ is not a good candidate for the hydrogenation of acetic acid in our tandem catalytic system even at lower reaction temperatures.

We have added the following sentence in the revised manuscript to briefly mention this result: “*The ethanol selectivity became lower at lower temperatures in the case of Cu/SiO₂ because of the decreased efficiency of acetic acid hydrogenation (Supplementary Table 3)*” (please see Page 8, Lines 1-3; Page 8, Lines 157-159 in the created PDF file).

Comment 6: Too many conclusions in the last part (line 240-263). This manuscript does not seem to have so many innovations. It needs to be more concise.

Reply and actions taken: Following the suggestion by the reviewer, we have shortened the conclusion of the revised manuscript. We have deleted the following sentences in the previous version of manuscript: “For syngas to methanol, ZnO-ZrO₂ solid solution is a highly selective catalyst. The acidity of metal oxides is responsible for the dehydration of methanol to dimethyl ether, which would produce methyl acetate, an undesirable product, instead of acetic acid in the subsequent carbonylation. The acidity of metal oxides should be eliminated, and thus the K⁺ modification is a simple and useful strategy. For methanol carbonylation, the Brønsted acid site in the 8-MR channels of H-MOR is the active site. The selective dealumination is crucial to keeping high selectivity and stability. For acetic acid hydrogenation to ethanol, Pt-Sn/SiC is a highly active and selective catalyst”. We believe that the following sentence is adequate to convey the key message: “*We have demonstrated that to keep the high selectivity of each step by carefully designing the corresponding catalyst is the key to obtaining high ethanol selectivity*” (please see Page 17, Paragraph 2, Lines 8-9; Page 18, Lines 405-407 in the created PDF file).

The following brief sentence has been added in the revised manuscript to describe the performance at a relatively higher CO conversion: “*At a CO conversion of ~10%, the ethanol selectivity could be sustained at 64%*” (please see Page 17, Paragraph 2, Lines 6-7; Pages 17-18, Lines 403-404 in the created PDF file). We have also added or modified the following brief sentences on the interplay between

different steps and compatibility, which are the most important issues for our tandem catalysis: “*The interplay between different steps and the compatibility of catalysts in syngas stream are also crucial. Methanol synthesis is the rate-determining step and the keeping of a sufficiently high CO/CH₃OH ratio plays a pivotal role in methanol-carbonylation step, which determines the C-C coupling. The presence of H₂ improves the stability of zeolite catalyst for methanol carbonylation, whereas the co-existence of CO requires the careful design of catalysts for hydrogenation of acetic acid to avoid the poisoning by CO*” (please see **Page 17, Paragraph 2, Lines 10-15; Page 18, Lines 407-413 in the created PDF file**).

Comment 7: In the Method part (line 292-297), “The resulting solid was subsequently calcined at 873 K in air for 5 h to remove pyridine molecules in the 12-MR channels to obtain an H-Na-MOR sample. The steam treatment of the H-Na-MOR was conducted at 723 K for 5 h in the quartz reactor with a stream containing 26% H₂O in N₂. The dealumination occurred selectively for the Al sites in the 12-MR channels”. Despite the support of the results of FT-IR and previous works, I am very suspicious that selective dealumination of the 12-MR channels can be really achieved. On the one hand, Na⁺ will migrate at high temperatures; on the other hand, some aluminum atoms are located at the intersection of the 8-MR channels and 12-MR channels. More evidences are needed to support it.

Reply and actions taken: The steam-treatment method for selective dealumination of framework Al sites in 12-MR of H-MOR zeolite was reported by Shen and co-workers (Ref. 42: *Catal. Commun.* **37**, 75-79 (2013)). The usefulness of this method was also confirmed in our previous work (Ref. 30: *Angew. Chem. Int. Ed.* **57**, 12012-12016 (2018)). By this method, a large part of Al species in 12-MR could be removed and the Al species in 8-MR channels could mainly remain intact. We agree with the reviewer that, in addition to the FT-IR, other characterization techniques are needed to further confirm this. Thus, we have further performed ²³Na MAS NMR, ²⁷Al MAS NMR, ¹H MAS NMR and pyridine-adsorbed FT-IR measurements.

Our partially dealuminated sample, H-MOR-DA-12MR, was prepared by exchanging of Na⁺ into pyridine-adsorbed H-MOR, the removal of pyridine at 873 K and subsequent steam treatment at 723 K, followed by exchanging with NH₄⁺ and calcination to remove Na⁺. It is expected that Na⁺ is mostly located in 8-MR channels, where pyridine cannot enter. The Al species with Na⁺ in the ion-exchanging position are more robust toward steam treatment, and thus, the Al species in the 12-MR channel are more prone to dealumination during steam treatment. The ²³Na MAS NMR measurement can discriminate Na⁺ cations in 12-MR and 8-MR channels of MOR because of their different chemical shifts (Ref. 45: Hunger, M., et al., *Solid*

State Nucl. Mag. **9**, 115-120 (1997)). Our ^{23}Na MAS NMR result showed that Na^+ cations were mainly located in 8-MR channels and remained stable during steam treatment at 723 K (Supplementary Fig. 4). The ^{27}Al MAS NMR can provide the information on the coordination of aluminium species in zeolite. The tetrahedrally coordinated framework Al gives a signal at 54 ppm, while the signal at 0 ppm corresponds to the octahedrally coordinated extra-framework Al. Our result showed that the signal at 54 ppm decreased for ~35% after the dealumination, indicating the removal of framework Al species by steam treatment (Supplementary Fig. 5). The ^1H MAS NMR result further demonstrated that the protons of Brønsted acid sites associated with framework Al species were partially removed after the dealumination (Fig. 3e). However, the ^{27}Al MAS NMR and ^1H MAS NMR cannot distinguish the Al species or protons in 12-MR or 8-MR channels. On the other hand, the pyridine-adsorbed FT-IR spectroscopy is able to provide insights into acid sites in different locations because pyridine can only enter the 12-MR channels of H-MOR. Our result showed significant decreases in the intensities of IR bands ascribed to acid sites, suggesting the occurrence of the dealumination in 12-MR channels of MOR by steam treatment (Supplementary Fig. 3b). Moreover, as already displayed in the previous version of manuscript, the FT-IR spectra in the range of 3540-3640 cm^{-1} provide information on OH groups in different locations including those in 8-MR and 12-MR (Refs. 48 and 49: Lukyanov D. B., et al. *J. Phys. Chem. C* **118**, 23918-23929 (2014); Cherkasov N., et al. *Vib. Spectrosc.* **83**, 170-179 (2016)). Combined with ^1H MAS NMR result that can provide the information about the total amount of Brønsted acid sites, we have estimated the amounts of Brønsted acid sites in 8-MR and 12-MR as well as those in the intersection between 8-MR and 12-MR channels through deconvolution of IR bands belonging to OH groups. The result has been shown in Fig. 3f and Supplementary Table 7. About 89% of Brønsted acid sites in 8-MR were retained, whereas 57% of Brønsted acid sites in 12-MR were removed after dealumination.

These new results have been added in the revised manuscript. The ^{23}Na MAS NMR ^{27}Al MAS NMR spectra have been added in Supplementary Figs. 4 and 5, while the pyridine-adsorbed FT-IR has been added in Supplementary Fig. 3b. The ^1H MAS NMR and FT-IR spectra of OH groups have been shown in Figs. 3e and 3f. Supplementary Table 4 in the previous version of manuscript has been modified to Supplementary Table 7. The paragraph to describe the characterization of the dealuminated MOR (please see the old manuscript, Page 8, Paragraph 3) has been completely rewritten as follows: “*The H-MOR-DA-12MR has been characterized by ^{23}Na , ^{27}Al and ^1H magic spinning nuclear magnetic resonance (MAS NMR) spectroscopic studies. Our dealuminated zeolite was prepared by exchanging of Na^+*

into pyridine-adsorbed H-MOR, removal of pyridine at 873 K and subsequent steam treatment at 723 K, followed by ion-exchanging with NH_4^+ and calcination to remove Na^+ cations⁴². Both XRD (Supplementary Fig. 3a) and TEM (Figs. 3c and 3d) results confirmed that the dealumination did not change the crystalline structure and morphology. The exchanged Na^+ cations are expected to be mainly located in the 8-MR channel, where pyridine cannot enter. This was confirmed by ^{23}Na MAS NMR measurements, which could discriminate Na^+ cations in different locations,⁴⁵ and the result showed that Na^+ mainly remained in 8-MR channels after steam treatment at 723 K (Supplementary Fig. 4). The Al species with Na^+ in the ion-exchanging position are more robust toward the steam treatment, and thus the Al species in the 12-MR channel are more prone to dealumination during steam treatment. ^{27}Al (Supplementary Fig. 5) and ^1H MAS NMR (Fig. 3e) results suggest the removal of framework Al species and Brønsted acid sites (Fig. 3e)^{46,47}. Pyridine-adsorbed FT-IR studies showed significant decreases in the intensities of IR bands ascribed to acid sites after steam treatment (Supplementary Fig. 3b), suggesting the occurrence of dealumination in 12-MR channels, because pyridine can only enter the 12-MR channel. FT-IR measurements in the range of 3540-3640 cm^{-1} also confirmed the partial removal of OH groups related to Brønsted acid sites in the 12-MR channel (Fig. 3f). The combination of the ^1H MAS NMR result, which could provide information of the total amount of Brønsted acid sites, and the deconvolution result of IR bands belonging to OH groups allowed us to estimate the amounts of Brønsted acid sites in 8-MR and 12-MR channels as well as in the intersection between the two channels^{48,49}. After dealumination, about 57% of Brønsted acid sites in 12-MR channels were removed, whereas 89% of Brønsted acid sites in 8-MR channels were retained (Supplementary Table 7). The selective removal of Brønsted acid sites in 12-MR channels should be the main reason for the enhancement in acetic acid selectivity by using the H-MOR-DA-12MR (Fig. 2b)” **(Please see from Page 10-Line 1 to Page 11-Line 1; Pages 10-11, Lines 209-235 in the created PDF file).**

In addition, we have also added the experimental details for ^{27}Al MAS NMR, ^1H MAS NMR, ^{23}Na MAS NMR and pyridine-adsorbed FT-IR in Catalyst characterization in Methods section in the revised manuscript **(please see Pages 21-22; Page 22 in the created PDF file).**

With all these new experimental results, we believe that we have provided adequate evidence for the location of Na^+ in 8-MR by exchanging Na^+ into the pyridine-adsorbed MOR and further evidence for the selective removal of framework Al species or Brønsted acid sites in 12-MR channels.

Response to Reviewer 3

General comment: The authors have studied the conversion of syngas to ethanol using relay-type catalysis with a sandwiched fixed-bed structure. They report a maximum selectivity of 90% at otherwise low overall CO conversion < 1%. The authors also provide some characterization data of their catalysts using standard methods like XRD, NH₃-TPD, TEM etc, and end up with some catalyst compatibility studies.

I am afraid to say that this paper does not provide the necessary breakthrough warranting publication in Nature Communications. The progress made with regard to their paper in “Angewandte” (Wei Zhou et al. *Angew. Chem. Int. Ed.* 2018, 57, 12012) is marginal. Furthermore, the present paper does not provide anything conceptually new since sandwiched layers of catalysts, to some extent similar to those used here, were previously used in their work. Of course, ZnO-ZrO₂ (known to be active in methanol production) was promoted with K⁺ and a Pt/Sn catalyst was added to create a triple bed catalyst structure, but does this provide sufficient material to go for a Nature Communications paper? Increasing the ethanol selectivity from 52%, as previously reported, to 90 % (or 81%) while encountering a decrease in the CO conversion from 6% to 0.6% (4%) just by replacing ZnAl₂O₄|H-Mor|ZnAl₂O₄ by ZnO-ZrO₂|H-Mor|Pt-Sn is simply not convincing. Finally, not much progress has been made with regard to a mechanistic understanding of the catalytic reactions.

I would recommend the authors provide a full story of their work (including advanced catalyst characterization and kinetic evidence) and submit to a journal devoted to catalysis.

Reply and actions taken: We appreciate the critical comments raised by this reviewer. As mentioned by the reviewer, the concept of bifunctional catalysis or relay catalysis for syngas conversions has been proposed in several of our previous papers (please see Refs. 26-30: *Angew. Chem. Int. Ed.* 55, 4725-4728 (2016); *Chem* 3, 334-347 (2017); *Chem. Sci.* 9, 4708-4718 (2018); *Angew. Chem. Int. Ed.* 57, 12012-12016 (2018)). This concept has become a powerful strategy in C1 chemistry and a lot of recent publications have been devoted to the conversion of syngas or hydrogenation of CO₂ to C₂₊ products following this strategy (please see our recent review paper that has been added as Ref. 2 in the revised manuscript, *Chem. Soc. Rev.* 48, 3193-3228 (2019)). However, almost all of the reported papers are devoted to the synthesis of hydrocarbons using this strategy. Only Ref. 30 (*Angew. Chem. Int. Ed.* 57, 12012-12016 (2018)) published by our group is devoted to the conversion of syngas to oxygenates. As pointed out by the reviewer, Ref. 30 also mentioned the formation of ethanol. However, Ref. 30 mainly reported the synthesis of methyl acetate from syngas via dimethyl ether (DME) intermediate. Although the formation of ethanol had also been mentioned, the selectivity of ethanol was ~50%. No detailed studies and no

effort to improve the performance have been carried out for the synthesis of ethanol in that work.

Ethanol is more versatile than methyl acetate. It is not only an important bulk chemical but also an ideal fuel additive. The direct synthesis of ethanol from syngas is one of the most attractive but very challenging targets in catalysis. The selectivity of ethanol reported to date typically cannot exceed ~60% even at limited CO conversions. The present work contributes to developing new catalytic systems for selective synthesis of ethanol using the tandem-catalysis strategy. We succeed in achieving ethanol selectivities of ~90% and 81% at CO conversions of 0.7% and 4.0%, respectively. At a CO conversion of ~10%, the ethanol selectivity can be sustained at 64%. Furthermore, we have demonstrated that the use of methanol carbonylation with CO in syngas to control C-C coupling is promising for the conversion of syngas to C₂ oxygenates. This work provides evidence for the crucial roles of the interplay between different steps and the compatibility of catalysts in syngas streams, offering guidance for the design of efficient tandem catalytic systems in C1 chemistry. The significant results and insights are further summarized as follows:

- (1) The tandem catalytic system composed of K⁺-ZnO-ZrO₂|H-MOR-DA-12MR|Pt-Sn/SiC has been designed and developed for the conversion of syngas to ethanol via methanol and acetic acid intermediates. Although the previous work has proposed a relay-catalysis route via DME and methyl acetate, the selectivity of ethanol is limited to 67% (actually ~50%). This is the first work to demonstrate the single-pass conversion of syngas to ethanol via methanol and acetic acid intermediates. The selectivities of ethanol reached 90%, 81%, 71% and 64% at CO conversions of 0.9%, 4.0%, 7.0% and 9.7%, respectively. The further comparison with other catalysts or catalytic systems shows significant advances in ethanol synthesis from syngas (Supplementary Table 1).
- (2) The key factors that affect the functions of each catalyst component have been clarified. The decrease in the Lewis acidity to suppress the dehydration to DME is the key to methanol selectivity and the K⁺-promoted ZnO-ZrO₂ solid solution is a suitable candidate. Steam-treated H-MOR catalyses the carbonylation of methanol to acetic acid rather than methyl acetate in the second step. The selective dealumination of H-MOR to remove Al species in 12-MR channels is crucial to the carbonylation. Pt-Sn/SiC shows a high activity for the hydrogenation of acetic acid to ethanol without hydrogenation of CO.
- (3) The interplay between different steps and the compatibility of catalysts with syngas stream have been elucidated. The carbonylation of methanol with CO

involves C-C coupling and is a key to the whole tandem catalysis. We discovered that the ratio of CO to methanol (CO/CH₃OH) should be kept sufficiently high to ensure the efficiency of the carbonylation reaction and this value also depends on the reaction temperature. Otherwise, the conversion of methanol to hydrocarbons (MTH) would proceed preferentially. Thus, the CO conversion in the first step should be controlled. For example, the CO/CH₃OH ratio should be ≥ 8.5 for methanol carbonylation at 583 K, and this requires the control of the CO conversion in the first step. We have further demonstrated that the stability of zeolite catalyst in methanol carbonylation is improved in syngas atmosphere because the presence of H₂ may avoid the carbon deposition. On the other hand, the design of catalysts for acetic acid hydrogenation should consider the effect of CO present in syngas. We found that the modification of Pt by Sn could eliminate the poisoning effect of CO by weakening the chemisorption of CO on Pt surfaces, contributing to the formation of ethanol.

In short, the novelty of the present work lies in the development of new catalytic system for ethanol synthesis with outstanding selectivity by designing tandem reactions and catalysts, and the elucidation of the chemistry of this tandem catalytic system, in particular the interplay and compatibility among the designed reaction steps and catalysts. The encouraging results and new insights gained in this work would guide the design of efficient tandem catalytic systems not only for syngas but also for the transformation of other C1 molecules (such as CO₂), and deepen our knowledge on the chemistry of tandem catalysis with interplay of multi-steps as well as compatibility of catalysts, thus providing a new opportunity to make breakthroughs in C1 chemistry. Therefore, we believe that this manuscript will attract wide attention in the research field of C1 chemistry, a hot research field under the background of utilization of non-petroleum carbon resources (natural gas or shale gas, coal, biomass and CO₂) to supply energy and chemicals, and thus is suitable for *Nature Communications*.

Response to Reviewer 4

General comment: In this manuscript, Wang et al. have shown the selective conversion of syngas to ethanol through a tandem process with three catalytic components. The results present in this work can be of general interests to the catalysis community, though the CO conversion is not high. The manuscript is well structured and written. This paper can be accepted by Nature Communications after revision.

Reply: We thank this reviewer for the positive comments on our manuscript. Our replies to the comments raised by the reviewer and the corresponding revisions are described as follows.

Comment 1: The characterization of PtSn/SiC show that part of the Pt species are positively charged (see Fig. S6). However, the XPS measurement was not carried out under in situ conditions. I suggest the authors to reduce the PtSn/SiC sample by H₂ and then transfer to the XPS analysis chamber to avoid contact with air. Considering the reaction conditions for syngas conversion, it is possible that Pt could be completely reduced under the reaction conditions.

Reply and actions taken: We appreciate this comment raised by the reviewer. Actually, the previous XPS measurements for the Pt/SiC and Pt-Sn/SiC samples, which were prepared by NaBH₄ reduction, were performed after treatment in N₂ atmosphere at 673 K in a pretreatment chamber directly connected to the detecting chamber, and the sample was transferred to the detecting chamber after the pretreatment without exposure to air. We have added these details in the Method section in the revised manuscript. Following the request by the reviewer, we have further performed XPS measurements for the Pt/SiC and Pt-Sn/SiC samples pretreated in syngas with H₂/CO of 1 at 583 K. Both Pt 4f and Sn 3d_{5/2} spectra for the Pt-Sn/SiC catalyst after syngas pretreatment have been displayed in Supplementary Figs. 9a and 9b. Similar XPS spectra were obtained for the samples after N₂ or syngas pretreatment.

Generally, Pt 4f_{7/2} with the binding energies of ~71.0 and 72.0-73.5 eV can be assigned to Pt⁰ and Pt²⁺, respectively (Refs. 50 and 51: Bera, P. et al., *Chem. Mater.* **15**, 2049-2060 (2003); de Miguel, S. R. et al., *J. Catal.* **184**, 514-525 (1999)). We observed binding energies of Pt 4f_{7/2} at 71.0 and 71.4 eV for the Pt/SiC and Pt-Sn/SiC catalysts after syngas pretreatment, respectively. These results suggest that Pt species on the Pt/SiC is in metallic state, while Pt^{δ+} also exists on the Pt-Sn/SiC catalyst. Regarding the oxidation state of tin, our XPS results reveal that the tin species on the Pt-Sn/SiC catalyst are mainly in cationic state (Sn²⁺ and/or Sn⁴⁺) with a Sn 3d_{5/2} binding energy at 487.4 eV. The deconvolution of Sn 3d_{5/2} peak suggests that a small fraction of metallic Sn species with a Sn 3d_{5/2} binding energy at 486.0 eV also appeared on catalyst surfaces.

Based on the results described above, we have rewritten the sentences describing the XPS results as follows: “*Quasi in situ X-ray photoelectron spectroscopy (XPS) studies showed that the binding energy of Pt 4f_{7/2} was 70.9-71.0 eV for the Pt/SiC catalyst after either N₂ pretreatment at 673 K or syngas (H₂/CO = 1) pretreatment at 583 K, indicating that Pt on this catalyst was in metallic state (Supplementary Fig.*

9a)^{50,51}. The binding energy of Pt 4f_{7/2} shifted to 71.4-71.5 eV for the Pt-Sn/SiC catalyst, suggesting that Pt^{δ+} also existed in the presence of Sn. The broad Sn 3d spectrum for the Pt-Sn/SiC catalyst after either N₂ or syngas pretreatment could be assigned mainly to cationic Sn species (Sn²⁺ and/or Sn⁴⁺) (Supplementary Fig. 9b)⁵¹. A small fraction of Sn⁰ also appeared on this catalyst” (please see from Page 11, Paragraph 2, Lines 6-13; Page 11, Lines 241-249 in the created PDF file). The following sentences to describe the experimental detail for XPS measurements have been added in the revised manuscript: “X-ray photoelectron spectroscopy (XPS) measurements were carried out in an UHV chamber equipped with an Omicron XPS (base pressure 5 × 10⁻¹⁰ torr). A monochromatized Al K_α X-ray source and a Sphere 2 analyser were used. After pretreated in N₂ at 673 K or syngas with H₂/CO = 1 at 583 K for 1 h in a pretreatment chamber directly connected to the detecting chamber, the sample was transferred into the UHV detecting chamber for XPS measurements” (please see Page 21, Lines 3-8; Page 21, Lines 506-511 in the created PDF file).

Comment 2: In Figure S7, the authors show the catalytic results obtained at different temperature. It is noted that, the CO conversion almost remain unchanged when increasing the temperature from 573 to 643 K. In a recent paper (Wang et al., Sci. Adv. 2017, 3: e1701290), the CO₂ conversion will increase sharply with the reaction temperature on ZnZrOx catalyst. Can the author give some explanation on the results shown in Figure S7?

Reply and actions taken: Methanol synthesis from syngas is a thermodynamically unfeasible reaction at higher temperatures. As displayed in Supplementary Fig. 1 in the revised manuscript, our thermodynamic calculations show that $\Delta_r G$ for methanol synthesis increases significantly with reaction temperature. The equilibrium conversion of CO becomes ~5% at > 600 K at H₂/CO = 1 (Supplementary Fig. 1b). Therefore, CO conversion over our K⁺-ZnO-ZrO₂ catalyst did not increase significantly with temperature at >600 K (Supplementary Fig. 10 in the revised manuscript) because of thermodynamic limitation, although it should increase kinetically.

The hydrogenation of CO₂ to methanol is also thermodynamically limited at high temperatures. As mentioned by the reviewer, Wang et al. reported that CO₂ conversion still increased significantly at high temperatures (> 613 K) (J. Wang, et al., Sci. Adv. 3: e1701290 (2017), Fig. 1 and Fig. S17a). However, it is noteworthy that the major product shifted from methanol to CO, whose selectivity reached 50-95%. In other words, the reverse water-gas shift (RWGS) reaction (CO₂ + H₂ → CO + H₂O), which is thermodynamically feasible under these conditions, became the major reaction. For the conversion of syngas to methanol, CO conversion may also increase

with temperature at > 600 K, but the formation of other products such as methane, which is thermodynamically feasible, dominates.

We have added the following sentence in the revised manuscript to explain this point: “*The CO conversion over this catalyst did not increase significantly at high temperatures due to the thermodynamic limitation (Supplementary Fig. 1)*” (please see Page 11, Paragraph 3, Lines 6-7; Page 12, Lines 257-259 in the created PDF file).

Comment 3: Since the CO conversion is not very sensitive to the reaction temperature. Can the CO conversion be improved by increasing the amount of K-ZnO-ZrO_x catalyst?

Reply and actions taken: We thank the reviewer for this constructive comment. We have investigated the effect of the amount of K⁺-ZnO-ZrO₂ in the combination of K⁺-ZnO-ZrO₂|H-MOR-DA-12MR|Pt-Sn/SiC on catalytic performances for syngas conversion at 583 K. The result has been added in Supplementary Table 6 in the revised manuscript. When the amount of K⁺-ZnO-ZrO₂ was increased from 0.66 g (typically used in this work) to 1.00 g, the CO conversion increased from 5.7% to 8.4% and the ethanol selectivity decreased slightly from 70% to 68%. However, a further increase in the amount of K⁺-ZnO-ZrO₂ to 1.50 g significantly decreased the ethanol selectivity to 59% due to the formation of C₁-C₄ hydrocarbons, although CO conversion still increased to 9.0%. At 1.50 g of K⁺-ZnO-ZrO₂, the increase in the amount of H-MOR-DA-12MR simultaneously to 1.00 g could slightly enhance the ethanol selectivity, and thus we could obtain a 9.7% CO conversion at ethanol selectivity of 64%.

We have added the following sentences in the revised manuscript to describe these results in the revised manuscript: “*The increase in the amount of K⁺-ZnO-ZrO₂ from 0.66 to 1.00 g in the combination of K⁺-ZnO-ZrO₂|H-MOR-DA-12MR|Pt-Sn/SiC increased CO conversion from 5.7% to 8.4% and the ethanol selectivity decreased only slightly to 68% at 583 K (Supplementary Table 6). A further increase in the amount of K⁺-ZnO-ZrO₂ to 1.50 g significantly decreased the ethanol selectivity*” (please see Page 8, Paragraph 2, Lines 2-6; Page 8, Lines 173-177 in the created PDF file).

Comment 4: In the triple tandem catalytic system, which step is the rate-limiting step? The CO hydrogenation on K-ZnO-ZrO_x or the carbonylation or the hydrogenation of acetic acid on PtSn/SiC? Some kinetic studies will help to answer this question and bring more insights on the reaction mechanism on the triple tandem system.

Reply and actions taken: We thank the reviewer for this constructive comment. Although we have investigated the effects of several kinetic factors such as temperature, H₂/CO ratio and total pressure on catalytic behaviors in the previous version of manuscript, we agree with reviewer that these results could not provide clear image of the whole reaction kinetics. Therefore, we have performed more detailed kinetic studies with our triple tandem catalytic system or the component for syngas conversions.

First, we have measured the catalytic behaviours of each catalyst component in each separated reaction at different reaction temperatures under the conditions. From the Arrhenius plot (Supplementary Fig. 11), we have calculated the apparent activation energies for syngas to methanol over K⁺-ZnO-ZrO₂, methanol carbonylation over H-MOR-DA-12MR, and acetic acid hydrogenation over Pt-Sn/SiC, and the results are 81, 71 and 45 kJ mol⁻¹, respectively. The rate of methanol synthesis is the lowest among the three steps in the same temperature range (503-583 K). For example, the formation rates of methanol (r_{MeOH}), acetic acid (r_{AA}) and ethanol (r_{EtOH}) for the three steps are 1.5, 6.7 and 21 mmol g⁻¹ h⁻¹, respectively, at 583 K and syngas pressure of 5 MPa. These results allow us to suggest that the methanol synthesis on the K⁺-ZnO-ZrO₂ is the rate-determining step in the tandem catalysis for syngas to ethanol.

Further, we have investigated the effects of partial pressures of CO and H₂ on catalytic behaviours at 583 K and the results have been shown in Fig. 4e and 4f. The rate of CO conversion depends strongly on the partial pressures of CO and H₂. Further analyses reveal that the experimental data could be fitted to a Langmuir-Hinshelwood kinetic model for the reaction between adsorbed CO and dissociatively adsorbed H species (Supplementary Figure 12). This kinetic behaviour is similar to that reported for the conversion of syngas to lower olefins over ZnO-ZrO₂/SSZ-13, where methanol synthesis on ZnO-ZrO₂ is the rate-determining step (Ref. 29: X. Liu, et al., *Chem. Sci.* **9**, 4078 (2018)). The result here further suggests that syngas to methanol is the rate-determining step in the tandem catalytic system. The product selectivity was also affected by the partial pressures of CO and H₂. A higher pressure of CO favoured the formation of ethanol, whereas a higher pressure of H₂ was unbeneficial to the ethanol selectivity, similar to the trends observed for changing the H₂/CO ratio.

Based on the additional kinetic results described above, we have rewritten the section of “Key reaction factors, stability and compatibility” by separating this section into two sections, i.e., “Reaction kinetics and stability” and “Interplay between different steps and compatibility” in the revised manuscript. The following two new paragraphs have been added to describe the new kinetic results described above: (1) “*We further investigated the temperature effect on catalytic behaviours of each*

catalyst component in each separated reaction. From the Arrhenius plot, the apparent activation energies for methanol synthesis, methanol carbonylation and acetic acid hydrogenation were calculated to be 81, 71 and 45 kJ mol⁻¹ over the K⁺-ZnO-ZrO₂, H-MOR-DA-12MR and Pt-Sn/SiC catalysts, respectively (Supplementary Fig. 11). The rate of methanol synthesis is the lowest among the three steps in a temperature range of 503-583 K. These results allow us to suggest that methanol synthesis on the K⁺-ZnO-ZrO₂ is the rate-determining step in the tandem catalysis for syngas to ethanol” (please see Page 12, Paragraph 2; Page 12, Lines 265-272 in the created PDF file); (2) “The effects of partial pressures of CO and H₂, denoted as P(CO) and P(H₂), on syngas conversions with K⁺-ZnO-ZrO₂|H-MOR-DA-12MR|Pt-Sn/SiC at 583 K are displayed in Fig. 4e and f. The rate of CO conversion, r(CO), depended strongly on both P(CO) and P(H₂). Further analyses reveal that the experimental data could be fitted to a Langmuir-Hinshelwood kinetic model for the reaction between adsorbed CO and dissociatively adsorbed H species (Supplementary Figure 12). This kinetic behaviour is similar to that reported for the conversion of syngas to lower olefins over a ZnO-ZrO₂/SSZ-13 catalyst, where methanol synthesis on ZnO-ZrO₂ is the rate-determining step²⁹. The result here further suggests that syngas to methanol is the rate-determining step in our tandem system. As expected, the product selectivity was also affected by P(CO) and P(H₂). A higher P(CO) favoured ethanol formation, whereas a higher P(H₂) was unbeneficial to ethanol selectivity, similar to the trends observed for changing the H₂/CO ratio” (please see Page 13, Paragraph 2; Pages 13-14, Lines 293-304 in the created PDF file).

Comment 5: How the mass ratio of the three components influence the activity and product distributions?

Reply and actions taken: We thank the reviewer for this important comment. Following this comment raised by the reviewer, we have investigated the effect of mass ratio of three components in the K⁺-ZnO-ZrO₂|H-MOR-DA-12MR|Pt-Sn/SiC combination on syngas conversions. The results have been added in Supplementary Table 6 in the revised manuscript.

First, by keeping the other two components at 0.66 g each, we changed the amount of K⁺-ZnO-ZrO₂. The increase in the amount of K⁺-ZnO-ZrO₂ from 0.66 to 1.00 g increased CO conversion from 5.7% to 8.4% and the ethanol selectivity decreased only slightly to 68% by at 583 K. A further increase in the amount of K⁺-ZnO-ZrO₂ to 1.50 g significantly decreased the ethanol selectivity. Then, we then changed the amount of H-MOR-DA-12MR by keeping the amounts of other two components. Both the CO conversion and ethanol selectivity slightly increased. Furthermore, we found that the amount of Pt-Sn/SiC did not affect CO conversion,

but a sufficient amount of Pt-Sn/SiC (≥ 0.55 g) was required to keep the ethanol selectivity at $\sim 70\%$.

We have added the following new paragraph to describe these results in the revised manuscript: “*The catalytic performance depends on the ratio of amounts of three catalyst components. The increase in the amount of K^+ -ZnO-ZrO₂ from 0.66 to 1.00 g in the combination of K^+ -ZnO-ZrO₂|H-MOR-DA-12MR|Pt-Sn/SiC increased CO conversion from 5.7% to 8.4% and the ethanol selectivity decreased only slightly to 68% at 583 K (Supplementary Table 6). A further increase in the amount of K^+ -ZnO-ZrO₂ to 1.50 g significantly decreased the ethanol selectivity. The increase in the amount of H-MOR-DA-12MR slightly improved the CO conversion and ethanol selectivity. The amount of Pt-Sn/SiC did not affect CO conversion, but a sufficient amount of Pt-Sn/SiC was required to keep a high ethanol selectivity. We obtained a 9.7% CO conversion at ethanol selectivity of 64%*” (please see **Page 8, Paragraph 2; Pages 8-9, Lines 172-181 in the created PDF file**).

Comment 6: what if the Pt loading in the Pt-Sn/SiC.

Reply and actions taken: The typical Pt loading used in this work is 1.0 wt%. We have further investigated the effect of Pt loadings on catalytic behaviours of syngas conversions using the K^+ -ZnO-ZrO₂|H-MOR-DA-12MR|Pt-Sn/SiC combination. The result has been shown in Supplementary Table 5 in the revised manuscript. We found that the ethanol selectivity increased slightly with an increase in Pt loading from 0.5 to 1.0 wt%, and a further increase in Pt loading had no effect on ethanol selectivity. CO conversion increased with an increase in Pt loading from 1.2 to 2.0 wt%, but the CO₂ selectivity increased, suggesting that an excess loading of Pt would promote the reverse water-gas shift (RWGS) reaction ($CO + H_2O \rightarrow CO_2 + H_2$). The lower olefins in the product were also mainly transformed into lower paraffins at high Pt loadings.

We have added the following sentence to describe the effect of Pt loading in the revised manuscript: “*The increase of Pt loading in the Pt-Sn/SiC from 0.5 to 1.0 wt% slightly increased ethanol selectivity, but a higher Pt loading (> 1.2 wt%) would promote the RWGS reaction to form more CO₂ and the hydrogenation of lower olefins to paraffins (Supplementary Table 5)*” (please see **Page 8, Lines 10-13; Page 8, Lines 166-169 in the created PDF file**). The following sentence has also been added in the Method section in the revised manuscript: “*The typical loadings of Pt and Sn were 1.0 and 1.2 wt%, respectively*” (please see **Page 19, Paragraph 2, Lines 8-9; Page 20, Lines 467-468 in the created PDF file**).

Comment 7: there are not details about pyridine IR study.

Reply and actions taken: We have performed pyridine FT-IR measurements for both metal oxides for methanol synthesis and zeolites for methanol carbonylation. The results have been displayed in Supplementary Figs. 2c and 3b. We have also added the following sentences to describe the procedure for measuring pyridine-adsorbed FT-IR in the Methods section in the revised manuscript: “*Pyridine adsorbed FT-IR measurements were performed with the same instrument. After pretreatment under vacuum at 673 K for 30 min, the sample was cooled down to 423 K. Then, pyridine adsorption was carried out at 423 K for 30 min. IR spectra were collected after gaseous or weakly adsorbed pyridine molecules were removed by evacuation at 423 K*” (please see Page 21, Lines 16-19; Page 22, Lines 519-522 in the created PDF file).

Moreover, the following sentences have been added in two places in the revised manuscript to describe the pyridine-adsorbed FT-IR results for metal oxides and for zeolites: “*Pyridine-adsorbed Fourier-transform infrared (FT-IR) measurements suggested that Lewis acid sites mainly existed on these metal oxides and the density of Lewis acid sites decreased in a trend similar to that from NH₃-TPD (Supplementary Fig. 2c)*” (please see Page 9, Paragraph 2, Lines 4-6; Page 9, Lines 199-202 in the created PDF file); “*Pyridine-adsorbed FT-IR studies showed significant decreases in the intensities of IR bands ascribed to acid sites after steam treatment (Supplementary Fig. 3b), suggesting the occurrence of dealumination in 12-MR channels, because pyridine can only enter the 12-MR channel*” (please see Page 10, Lines 14-17; Page 10, Lines 223-226 in the created PDF file).

Comment 8: the present results and catalysts (specially for syngas to acetic gas) should be compared and discussed from those presented in patents from BP company.

Reply and actions taken: Following this comment raised by the reviewer, we have searched the patents from BP company for the synthesis of acetic acid and found three related patents.

Two of these patents disclosed the single-step synthesis of C₂-oxygenates from syngas using Rh-based catalysts (Rh-Mn-Fe-M/SiO₂), over which the selectivity of C₂ oxygenates (including ethanol, acetaldehyde and acetic acid) was 50-60%, (Atkins, M. P. Process for the conversion of synthesis gas to oxygenate. *U. S. Patent*, No.: US7939571B2; Atkins, M. P. Process for the conversion of synthesis gas to oxygenate. *U. S. Patent*, No.: US8063110B2). As compared to these BP-based Rh-Mn-Fe-M/SiO₂ catalysts, our tandem system of K⁺-ZnO-ZrO₂|H-MOR-DA-12MR shows significantly higher selectivity of acetic acid (60-90%) (Supplementary Table 2). Although the major aim of our study is to synthesize ethanol rather than acetic acid, the bifunctional tandem system of K⁺-ZnO-ZrO₂|H-MOR-DA-12MR could catalyse the formation of acetic acid with

high selectivity.

The third related patent from BP reported an indirect multiple-process route for the production of acetic acid from syngas and the route consisted of multiple reaction zones and separation units (Bristow, T. C. Integrated process for making acetic acid from syngas. *Eur. Patent*, No.: EP2935184B1). This route integrates reactions of syngas to methanol, methanol to DME, DME carbonylation to methyl acetate and dehydration-hydrolysis reaction of methyl acetate with methanol to produce acetic acid. The estimation showed a ~50% selectivity of acetic acid.

We have added typical results of the three patents from BP company in Supplementary Table 2 in the revised manuscript. These patents have been listed as references 25-27 in Supplementary References. In the main text of the revised manuscript, we have modified the following sentence: “Rh-based catalysts have been exploited for the conversion of syngas into acetic acid (Supplementary Table 2). The present K^+ -ZnO-ZrO₂|H-MOR-DA-12MR system shows significantly higher acetic acid selectivity” to “*Rh-based catalysts as well as an indirect route integrated by multiple reaction processes via methanol, DME and methyl acetate have been exploited for the conversion of syngas into acetic acid (Supplementary Table 2). The present K^+ -ZnO-ZrO₂|H-MOR-DA-12MR system shows significantly higher acetic acid selectivity than most of the Rh-based catalysts or the multi-process route*” (***please see Page 7, Paragraph 1, the last two sentences; Page 7, Lines 146-150 in the created PDF file***).

Reviewers' comments:

Reviewer #1 (Remarks to the Author):

The research article by Kang et al. submitted to Nature Communications on triple tandem catalysts (K-ZnO-ZrO₂ | H-MOR-DA-12-MR | PtSn/SiC) for direct conversion of synthesis gas into ethanol has been improved significantly since the previous version of the manuscript. The added discussions on CO₂ production, thermodynamics and catalyst grain size have addressed most of my detailed comments on the manuscript.

However, after re-reading the manuscript and the comments and responses to the other reviewers, I believe that this manuscript does not possess the novelty or rigor to merit publication within Nature Communications. The manuscript is not a significant leap forward from the previous report published by the authors (Angew. Chem. Int. Ed. 57, 437 12012-12016 (2018)). There is also a lack of mechanistic insight that would be expected within a manuscript of this level. To iterate more clearly:

1. The authors response regarding thermodynamics could be more rigorous. For example, the point stating: 'The three catalyst components are separated with each other in one reactor, and thus the thermodynamics is different from that for the direct conversion of syngas to ethanol' is objectively incorrect. The thermodynamics for the overall reaction are fixed, regardless of the intermediates. It is important to note that the authors do provide details on the maximum achievable CO conversion and EtOH selectivity, which are only theoretically possible if each of the three steps is perfectly optimized, eg. high CO:MeOH ratio after methanol synthesis. Although this design constraint appears to maximize EtOH selectivity, the counterpoint is that the EtOH selectivity would be further maximized by running this process at three optimized temperatures/pressures in three different reactors, and perhaps feeding CO at different points in the reactor. Therefore, what is the advantage/breakthrough of a triple-tandem system?

2. The stated advantages of the triple-tandem catalyst (lower energy input, higher efficiencies, easier separation) are not well-supported, particularly because of the large amount of water formation and low selectivity toward EtOH, which would require an energy intensive separation post-reaction. Is there evidence that the triple tandem catalyst could replace fermentation for certain EtOH end-use applications?

3. The mechanistic questions regarding the catalyst need to be addressed. For example, hydrogen has extremely high dispersion, resulting in near constant H₂ concentration throughout the catalyst bed (J. Catal. 372, 2019, 370-381). The results obtained when changing the amount of quartz wool/mixing the catalyst in a dual bed are quite interesting, but what is their significance? Why is a distinct difference in selectivity observed in the different bed configurations? Can the authors offer any direct evidence other than the reported reactor data?

4. There is a comment in the reviewer's response stating 'Regarding the reproducibility, our experimental results are reproducible and the relative deviation is less than 5% in all cases.' If the authors have already performed the experiments, it would be useful to include these data in the form of error bars or within the SI in a future revision.

Overall, the manuscript does present some interesting findings, but is not a good fit for Nature Communications.

Reviewer #2 (Remarks to the Author):

The author has made a more comprehensive introduction and appropriate comments on the progress of this field in their revised manuscript. They have also added more comparative

experiments to prove the progress that they have made. In addition, more characterization evidence has been provided to support the structure of the catalysts. Therefore, in my opinion, this revised manuscript can be accepted.

Reviewer #5 (Remarks to the Author):

Multifunctional catalytic reactor operation is an important, clear new direction for bulk (petro)chemical industrial processes and hold promise for efficiency and energetic breakthroughs, eliminating intermediate separation processes and simplifying processing. Several examples have been given in literature for bifunctional systems, but in this manuscript three catalytic functions are combined in an operation where also thermodynamic constraints have to be faced, resulting in high product selectivity for an oxygenated product.

It is a new development compared with their former paper (ref 30), but with much lower selectivities and other products.

Here, the authors convincingly demonstrate the direct conversion of syngas mixtures into ethanol with high selectivity at a reasonable CO conversion in one reactor operation by a combination of three consecutive catalyst layers, each catalyzing cascade-wise methanol synthesis, methanol carbonylation and selective acetic acid hydrogenation. Mixing the catalysts resulting in more proximate/intimate catalytic functionalities did not work.

To operate in combination under similar conditions the three catalysts were successfully identified and optimized to be able to do so.

It must be realized for this case that the consecutive conversion process will still face the methanol synthesis thermodynamic limitation in the first step, i.e. limiting the CO conversion, but eliminates intermediate separation processing. On the other hand, the consecutive reaction requires a high CO/methanol ratio, so CO conversion cannot be too high to achieve a compatible operation, and a subtle balance between the performances of the catalysts had to be found in terms of catalyst properties, amounts and operating conditions. Limited syngas conversion is probably also the reason for the stable carbonylation performance where the hydrogen may remove or suppress coke deposition on the zeolite catalyst.

Of course, the limited CO conversion limits the yield of the ethanol product and separation of the reactor product mixture is still required. Some directions for further developments are mentioned.

The main manuscript is accompanied by extensive supporting information on catalyst characterization, composition and conditions dependencies, including new data. The role/function of the individual catalytic components is clearly demonstrated by the presented data.

Regarding this revision of the original manuscript, for which the authors have performed additional experimental work, they addressed the questions of the reviewers in a very adequate manner. Together with the various advanced developments presented in this work (triple tandem operation for one-reactor syngas conversion into ethanol, subtle fine-tuned catalysts' optimization, compatible stable catalyst operation) I can only recommend acceptance of this manuscript.

Responses to Reviewers

Response to Reviewer 1

General comment: The research article by Kang et al. submitted to Nature Communications on triple tandem catalysts (K-ZnO-ZrO₂|H-MOR-DA-12-MR|PtSn/SiC) for direct conversion of synthesis gas into ethanol has been improved significantly since the previous version of the manuscript. The added discussions on CO₂ production, thermodynamics and catalyst grain size have addressed most of my detailed comments on the manuscript.

However, after re-reading the manuscript and the comments and responses to the other reviewers, I believe that this manuscript does not possess the novelty or rigor to merit publication within Nature Communications. The manuscript is not a significant leap forward from the previous report published by the authors (*Angew. Chem. Int. Ed.* 57, 437 12012-12016 (2018)). There is also a lack of mechanistic insight that would be expected within a manuscript of this level. To iterate more clearly:

Reply: We appreciate the critical comments raised by this reviewer. However, we cannot agree on his/her comment on significance and novelty of our present manuscript because of the following reasons.

In our previous paper, which was already cited as Ref. 31 (*Angew. Chem. Int. Ed.* 57, 12012-12016 (2018)) in our present manuscript, we reported that the relay catalysis by combining metal oxide and zeolite could be used for the conversion of syngas into different types of products, including methyl acetate, ethanol and ethylene via dimethyl ether (DME) intermediate. In that paper, we demonstrated that the combination of ZnAl₂O₄ and H-MOR could catalyse the formation of C₂ products, in particular methyl acetate with high selectivity > 85% via DME intermediate. We also revealed that the on-site removal of water, which was generated during DME formation and could poison zeolite for the subsequent carbonylation, by water-gas shift reaction was a key issue for the production of methyl acetate by relay catalysis. As pointed out by the reviewer, our previous paper also mentioned the formation of ethanol from syngas by relay catalysis, but ethanol formation is not the focus of that work. The selectivity of ethanol was only ~50% with a CO conversion of ~6%. We just showed the possibility of ethanol formation by relay catalysis in the previous paper without detailed studies and effort to improve the performance for ethanol formation.

The significance and novelty of the present manuscript lie in the following aspects:

First, the present manuscript focuses on the single-pass conversion of syngas into ethanol by tandem catalysis. Ethanol is a kind of very important and versatile product,

not only being an important bulk chemical but also an ideal fuel additive. The direct synthesis of ethanol from syngas is a very attractive but challenging goal in C1 chemistry. The selectivity of ethanol reported to date generally cannot exceed 60% even at limited CO conversions (Supplementary Table 1). The present work contributes to developing new multifunctional catalytic systems working in tandem for the selective conversion of syngas into ethanol. We have succeeded in single-pass conversion of syngas to ethanol with ethanol selectivities of 90%, 81% and 71% at CO conversions of 0.9%, 4.0% and 7.0%, respectively. At a CO conversion of ~10%, the ethanol selectivity could be sustained at 64%. These overall performances for syngas to ethanol are much higher than those reported in our previous paper and the paper by Zhang et al. (Ref. 32).

Second, the chemistry for the tandem-catalytic system developed in the present work is different from that reported in our previous work. As already mentioned, DME is a key intermediate for the formation of methyl acetate and further ethanol in the previous system, whereas methanol is designed as a key reaction intermediate in the present work. The carbonylation of methanol with CO existing in syngas produces acetic acid and ensures the precise C–C coupling in our system, and ethanol is subsequently hydrogenated to ethanol in the third step. This tandem route via methanol and acetic acid intermediates can ideally accomplish 100% selectivity of ethanol in single pass if highly selective catalysts can be developed for the three steps. On the other hand, the route via DME and methyl acetate can only provide a maximum ethanol selectivity of 67% in one pass. We have carefully designed highly selective catalyst for each step, i.e., $K^+ZnO-ZrO_2$ for methanol synthesis, H-MOR with Al species in 12-membered-ring (12-MR) channels selectively removed (denoted as H-MOR-DA-12MR) for methanol carbonylation and Pt-Sn/SiC for acetic acid hydrogenation. The combined triple tandem catalytic system has resulted in selective formation of ethanol. As mentioned above, we have achieved ethanol selectivities of 90%, 81% and 71% at CO conversions of 0.9%, 4.0% and 7.0%, respectively.

Third, our present work has not only clarified the structure and physicochemical property of each catalyst component that can lead to high selectivity for each step but also has elucidated the interplay between the tandem steps and the compatibility of catalyst component in syngas stream. *For the structure and physicochemical property of each catalyst component:* we uncovered that the oxygen vacancies on $K^+ZnO-ZrO_2$ play pivotal roles in the activation of CO and H_2 , while the acidity on oxide surfaces, which catalyses the dehydration of methanol to DME, is detrimental to methanol selectivity; the selective removal of Al species in the 12-MR channels of H-MOR is the key to obtaining high selectivity of acetic acid in the carbonylation of methanol; the modification of Pt/SiC with Sn to suppress the CO chemisorption is

crucial to the selective hydrogenation of acetic acid to ethanol. *For the interplay between the tandem steps and the compatibility of catalyst component in syngas:* we found that a sufficiently high CO/CH₃OH ratio, which is determined by the CO conversion in the first step, is required for the carbonylation of methanol, and thus is a key to ethanol selectivity; the stability of H-MOR-DA-12MR catalyst for methanol carbonylation is significantly improved in syngas stream as compared to that in the traditional CO stream; the catalyst for acetic acid hydrogenation should be carefully designed to avoid the poisoning effect of CO in syngas. These insights, which have not been touched in the previous publication, would significantly contribute to enrich the chemistry of the tandem catalysis for syngas conversions and to future designing more efficient catalysts for ethanol synthesis.

Further, we have performed kinetic studies for tandem catalytic system for the conversion of syngas to ethanol. Based on the measurements of reaction rate and activation energy for each step, we disclosed that methanol synthesis over the K⁺-ZnO-ZrO₂ is the rate-determining step in the tandem system. A Langmuir-Hinshelwood kinetic model was found to be suitable for describing the conversion of syngas. In other words, the syngas conversion takes place between adsorbed CO and dissociatively adsorbed H species (Supplementary Fig. 14).

In short, different from the previous paper (Ref. 31), our present manuscript aims at developing efficient triple tandem catalytic system for single-pass conversion of syngas to ethanol, one of the most important C₂ compounds, with high selectivity. Ethanol selectivities of 90%, 81% and 71% have been achieved at CO conversions of 0.9%, 4.0% and 7.0%, respectively. Such high overall performances have not been reported in previous publications. Unlike previous systems, where DME is the key intermediate, methanol and acetic acid are the intermediates in the present triple tandem system, and this enables high selective formation of ethanol in single pass in principle. Besides the importance of carefully designing each catalyst to ensure the high selectivity for each step, we have demonstrated the key roles of interplay between reaction steps and compatibility of catalysts in syngas stream. Reaction kinetics has also been performed to gain insights into the reaction mechanism. The encouraging results and new insights gained in this work would guide further design of efficient tandem catalytic systems not only for syngas conversion but also for the transformations of other C₁ molecules (such as CO₂ and CH₄). Thus, we believe that the present work provides a new opportunity to control product selectivity in the field of C₁ chemistry. The insights can also deepen our knowledge on the chemistry of tandem catalysis with interplay among multi-steps as well as compatibility of catalysts, and thus will attract broad attention. Therefore, we believe that our present manuscript is suitable for publication in *Nature Communications*.

Comment 1: The authors response regarding thermodynamics could be more rigorous. For example, the point stating: 'The three catalyst components are separated with each other in one reactor, and thus the thermodynamics is different from that for the direct conversion of syngas to ethanol' is objectively incorrect. The thermodynamics for the overall reaction are fixed, regardless of the intermediates. It is important to note that the authors do provide details on the maximum achievable CO conversion and EtOH selectivity, which are only theoretically possible if each of the three steps is perfectly optimized, eg. high CO:MeOH ratio after methanol synthesis. Although this design constraint appears to maximize EtOH selectivity, the counterpoint is that the EtOH selectivity would be further maximized by running this process at three optimized temperatures/pressures in three different reactors, and perhaps feeding CO at different points in the reactor. Therefore, what is the advantage/breakthrough of a triple-tandem system?

Reply and actions taken: The reviewer has pointed out that the thermodynamics for an overall reaction is fixed, regardless of the intermediates. This is certainly correct for the overall chemical reaction with all reactants, products and intermediates mixed together in one closed system. However, in our triple tandem system, although the three-step reactions are carried out in the same fixed-bed flow reactor, the three catalyst components (i.e., $K^+ZnO-ZrO_2$, H-MOR-DA-12MR and Pt-Sn/SiC) are packed sequentially and spatially isolated with quartz wool. We have demonstrated that the isolation of the three catalyst components can result in high ethanol selectivity. Thus, the three-step reactions actually take place in an independent manner, and the two intermediates, methanol and acetic acid, are formed sequentially at different catalyst beds without being mixed together. This is quite different from the case of the batch reactor, where all the reactants, products and intermediates in the tandem system are mixed together. Therefore, the thermodynamics of each step (i.e., syngas to methanol, methanol carbonylation and acetic acid hydrogenation) rather than that of the overall reaction (syngas to ethanol) should be considered.

To explain this point more clearly, we have modified or added the following sentences in the revised Supplementary Information in describing the thermodynamic analysis: *“The tandem catalysis is composed of three steps, i.e., methanol synthesis, methanol carbonylation and acetic acid hydrogenation, and the three catalyst components are separated with each other by quartz wool in one fixed-bed flow reactor. The reaction in each step occurs relatively independently over the corresponding catalyst bed in the reactor and the intermediates, i.e., methanol and acetic acid, cannot be mixed together. Thus, the thermodynamics of each step should*

be considered” (please see Supplementary Information, Page 4, section of Explanation of the calculations, Lines 1-7).

Regarding the advantage of the single-pass route, we think that, as compared to the indirect route composed of several different processes, the single-pass conversion of syngas to ethanol by triple-tandem catalysis is a simple and efficient method. The indirect route consists of multiple processes including syngas to methanol/DME, methanol/DME to acetic acid/methyl acetate and hydrogenation of acetic acid/methyl acetate to ethanol (Fig. 1, Routes B and C). These individual processes require separated sets of reaction equipment and the products in all the three processes need to be separated and purified. The single-pass conversion of syngas to ethanol in one reactor would simplify the reaction and separation/purification (only one separation/purification of ethanol from other by-products is required) processes as well as the required equipment, thus reducing the cost and improving the operation efficiency. We believe that this is the major reason for why a large number of studies have been devoted to the direct synthesis of ethanol from syngas (Fig. 1 Route A), although the progress has been very limited and the selectivity of ethanol can hardly exceed 60% even at limited CO conversions. We have achieved ethanol selectivities of 90%, 81%, 71% and 64% at CO conversions of 0.9%, 4.0%, 7.0% and 10%, respectively, by using the triple tandem catalytic system. Therefore, the single-pass triple tandem catalysis not only has the advantages of direct conversion route, i.e., simplified reaction and separation/purification processes as well as the required equipment, but also provides a new route (Fig. 1, Route D) that can achieve high ethanol selectivity.

Furthermore, we have found that the tandem catalysis in one reactor can benefit the performance of methanol carbonylation with zeolite H-MOR. H-MOR undergoes quick deactivation during the carbonylation of methanol in the conventional CO stream because of the carbon deposition, and this is known as a challenging issue (Ref. 55). We have demonstrated that the presence of H₂ in syngas steam significantly enhances the stability of zeolite catalyst probably because H₂ could suppress the carbon deposition (Fig. 5a and Supplementary Fig. 15). In addition, as compared to the high concentration of acetic acid in the indirect process, that may cause corrosion of reaction equipment, the rapid consumption of acetic acid in the subsequent step in the tandem catalysis would also be beneficial.

In short, our single-pass triple tandem catalysis not only has the advantages of the direct conversion route, i.e., simplified reaction and separation/purification processes as well as the required equipment, but also provides an efficient method to achieve high ethanol selectivity that significantly outperforms those reported for the direct conversion of syngas to ethanol. The tandem catalysis has also brought about

beneficial effect on the stability of zeolite catalyst for carbonylation reaction, which is a very challenging issue. Based on these results and discussion, we have added the following sentences in the revised manuscript: “*Furthermore, although ethanol can be produced through three separated processes, i.e., methanol/DME synthesis, methanol/DME carbonylation and acetic acid/methyl acetate hydrogenation (Fig. 1, Routes B and C), the single-pass route reported in this work would simplify the reaction and separation/purification processes as well as the required equipment, and can thus reduce the cost and improve the operation efficiency. The tandem catalysis in one reactor can also bring about beneficial effect on the stability of zeolite catalyst for carbonylation reaction, which is known as a challenging issue*⁵⁵” (please see Page 17, Paragraph 2, Lines 4-10).

Comment 2: The stated advantages of the triple-tandem catalyst (lower energy input, higher efficiencies, easier separation) are not well-supported, particularly because of the large amount of water formation and low selectivity toward EtOH, which would require an energy intensive separation post-reaction. Is there evidence that the triple tandem catalyst could replace fermentation for certain EtOH end-use applications?

Reply and actions taken: First, as described above, the indirect routes typically involves three individual processes, i.e., syngas to methanol/DME, methanol/DME to acetic acid/methyl acetate and hydrogenation of acetic acid/methyl acetate to ethanol (Fig. 1, Routes B and C). The separation and purification of products in all the three processes are required, leading to higher energy input in these separation and purification. In contrast, only one separation/purification of ethanol from liquid by-products is needed in the tandem catalytic route, and this would simplify the separation and purification process. The following sentences have been added in the revised manuscript to discuss this issue: “*Furthermore, although ethanol can be produced through three separated processes, i.e., methanol/DME synthesis, methanol/DME carbonylation and acetic acid/methyl acetate hydrogenation (Fig. 1, Routes B and C), the single-pass route reported in this work would simplify the reaction and separation/purification processes as well as the required equipment, and can thus reduce the cost and improve the operation efficiency. The tandem catalysis in one reactor can also bring about beneficial effect on the stability of zeolite catalyst for carbonylation reaction, which is known as a challenging issue*⁵⁵” (please see Page 17, Paragraph 2, Lines 4-10).

Second, regarding the formation of water in the synthesis of ethanol from syngas, no matter which routes you choose, direct or indirect routes, water will be formed along with ethanol, because of the decrease in O/C ratio from 1:1 for CO to 1:2 for ethanol. The amount of water formation increases with a decrease in the selectivity of

ethanol and an increase in the selectivity of hydrocarbons, which are the major byproducts, because all the oxygen atoms in CO should be removed as water in the case of hydrocarbon formation. Because the selectivity of ethanol in our present work is significantly higher than those reported for the direct conversion of syngas to ethanol (Supplementary Table 1), the formation of water is not as serious as in other routes. In this context, the reviewer's comment on the low selectivity of ethanol is not the case for our system. Nevertheless, we do agree with the reviewer that the separation of water and ethanol should be considered for future applications. Actually, we already added the following sentences in the previous revised manuscript: "*The separation of H₂O formed together with ethanol in the last step needs consideration to obtain anhydrous ethanol. The extractive distillation, a technique widely used for the separation of ethanol-water azeotropes derived from ethylene hydration and sugar fermentation⁵⁹, as well as membranes^{60,61} may fulfil this task*" (**please see Page 17, Paragraph 2, Lines 13-16 in the revised manuscript**).

Third, for the synthesis of ethanol by fermentation in batch or even in fed-batch mode, it is reported that the concentration of ethanol produced, in most cases, does not exceed 4-5% (w/w). This is because of the difficulty in working with high loading of dry solid reactant exceeding 15% (w/w) in the reactor (please see Ref. 6: Kennes, D. et al. *J. Chem. Technol. Biotechnol.* **91**, 304-317 (2016); Ref. 7: Shen, F. & Liu, R. *Energy Fuels* **23**, 519-525 (2009)), although the fed-batch mode usually allows for a higher production rate. Thus, the separation of ethanol with water needs high energy input. Actually, the high energy-demanding feature of the fermentation process has been pointed out in literature. For example, it is pointed out that "the mixture with a lower ethanol concentration made the distillation process very energy-intensive" (Bai, F. W. et al. *Biotech. Adv.* **26**, 89-105 (2008)); "The high energy demand of fermentation processes due to the distillation steps required to isolate the products and.....hampering a wider industrialization of these technologies" (Ref. 8: Luk, H. T. et al. *Chem. Soc. Rev.* **46**, 1358-1426 (2017)). In contrast, the ethanol content, in the ethanol/water mixture produced from syngas, is calculated to be 71% by taking into account that one mole of ethanol is formed concomitantly with one mole of water. Even if the selectivity of ethanol decreases to 64% (CO conversion is 10%) as in our work, the ethanol content in the ethanol/water mixture can also be maintained at 54.6%, which is much higher than that in the fermentation process. Because of the significant difference in the concentration of ethanol produced in the mixture of liquid products, the fermentation process is considered to be more energy-demanding for ethanol separation.

Nevertheless, we do not want to conclude that the synthesis of ethanol from syngas using the present triple tandem catalysis can replace the production of ethanol

by the fermentation process. The technology of the fermentation of sugars or starch to produce ethanol is quite mature. The fermentation method is quite suitable for the countries rich in crop production (like Brazil). However, the fermentation method is difficult to be implemented in countries not rich in crops. Instead, syngas can be produced from coal, natural or shale gas, biomass (that cannot undergo fermentation efficiently) and even CO₂. The purpose of our study is not to replace the fermentation process for ethanol production, but to diversify the method for ethanol production by utilizing various carbon resources. Because our present work is a fundamental study, we hope to present a promising route for ethanol synthesis via single-pass conversion of syngas.

Comment 3: The mechanistic questions regarding the catalyst need to be addressed. For example, hydrogen has extremely high dispersion, resulting in near constant H₂ concentration throughout the catalyst bed (*J. Catal.* 372, 2019, 370-381). The results obtained when changing the amount of quartz wool/mixing the catalyst in a dual bed are quite interesting, but what is their significance? Why is a distinct difference in selectivity observed in the different bed configurations? Can the authors offer any direct evidence other than the reported reactor data?

Reply and actions taken: We appreciate these comments raised by this reviewer. As already described, our triple tandem catalytic system is composed of three catalyst components, i.e., K⁺-ZnO-ZrO₂ for syngas to methanol, H-MOR-DA-12MR for methanol carbonylation and Pt-Sn/SiC for hydrogenation of acetic acid, and the three components are separated with each other by quartz wool. We have performed catalyst characterizations and discussed the mechanism on each catalyst component, and these have mainly been described in the section of “Structural characterizations of tandem catalysts”. We have also studied the proximity of catalyst components by changing granule sizes of catalyst components and the amount of quartz wool/mixing (in the section of “Interplay between different steps and compatibility”). These results enable us to emphasize that the complete separation of the three components is the key to the selective formation of ethanol. We think that these results and the related discussion are insightful for understanding the chemistry of the tandem catalysis for syngas conversions and are useful for future design of more efficient tandem catalytic systems. The novel insights are summarized as follows.

First, we studied the mechanism of each reaction on the corresponding catalyst surfaces. For methanol synthesis on K⁺-ZnO-ZrO₂, the activation of CO and H₂ is the key. Some studies have shown that Zr-based catalysts can work for CO or H₂ activation (Ref. 30: *Chem. Sci.* **9**, 4708-4718 (2018), cited as Ref. 29 in the previous version; Ref. 45: *J. Catal.* **118**, 400-416 (1989), cited as Ref. 44 in the previous

version; Ref. 47: *J. Catal.* **372**, 370-381 (2019), a new reference recommended by the reviewer). Based on the references (Refs. 30 and 45), we proposed in the previous version of manuscript that the oxygen vacancy sites on metal oxide surfaces may function for the activation of CO. The incorporation of ZnO into ZrO₂ to form solid solution could enhance the generation of oxygen vacancies, thus favoring the activation of CO. To strengthen this point, we have performed electron paramagnetic resonance (EPR) measurements to characterize the oxygen vacancies on metal oxides. Based on the results displayed in Supplementary Fig. 3, we have rewritten the paragraph for discussing the generation of oxygen vacancies and their roles in syngas conversion as follows: *“Oxygen vacancy sites on ZrO₂ surfaces have been proposed to account for CO activation⁴⁵. Our electron paramagnetic resonance (EPR) spectroscopy measurements showed a signal at g value of 2.003 (Supplementary Fig. 3), which could be attributed to oxygen vacancies over ZrO₂-based metal oxides⁴⁶. The incorporation of ZnO into ZrO₂ to form solid solution enhanced the signal at g value of 2.003 and thus the generation of oxygen vacancies over both ZnO–ZrO₂ and K⁺–ZnO–ZrO₂ samples. We believe that the increased density of oxygen vacancy sites favours the activation of CO as well as H₂^{30,45,47}, (please see from Page 9-Paragraph 3 to Page 10-Paragraph 1)*. We have also added the experimental description of ESR studies in the section of “Method-Catalyst characterization” (please see Page 21, Lines 5-2 from bottom). Furthermore, using NH₃-TPD and pyridine adsorbed FT-IR, we have revealed that the decrease in acidity of metal oxides is crucial to keeping methanol from further conversion to DME (please see Page 9, Paragraph 2). In addition, through kinetic studies, we have identified that methanol synthesis is the rate-determine step in the triple tandem catalysis and a Langmuir-Hinshelwood model could be used to describe the conversion of syngas to methanol (please see from Page 11-Paragraph 3 to Page 12-Paragraph 2 and Page 13-Paragraph 3). For the modified H-MOR-catalyzed carbonylation of methanol, through characterization using ²³Na, ²⁷Al and ¹H MAS NMR, pyridine adsorbed FT-IR and FT-IR spectra of OH groups, we have demonstrated that the Brønsted acid sites in 8-MR channels of H-MOR work for the selective carbonylation of methanol with CO to acetic acid (please see from Page 10-Paragraph 2 to Page 11-Paragraph 1). For the hydrogenation of acetic acid on Pt–Sn/SiC, we have demonstrated that the modification of Pt with Sn formed Pt–Sn alloy, which significantly decreases the chemisorption of CO, can avoid the poisoning effect of CO on hydrogenation reaction, thus accelerating the hydrogenation of acetic acid to ethanol (please see Page 11, Paragraph 2; Page 15, Paragraph 2).

Second, we investigated the effect of proximity (configuration) of the catalyst components in the triple tandem system on their catalytic behaviors for syngas

conversion. It should be noted that this is somewhat different from the study pointed out by the reviewer (Razdan, N. K. et al., *J. Catal.* **372**, 370-381 (2019)), where the configuration of metallic Zr (as H₂ absorption materials) and MoC_x/H-ZSM-5 (as catalyst) was studied to suppress axial H₂ partial pressure profiles and to enhance the net dehydrogenative aromatization rate. In our case, we only investigated one configuration, i.e., K⁺-ZnO-ZrO₂|H-MOR-DA-12MR|Pt-Sn/SiC placed layer-by-layer and separated by quartz wool in the reactor, and we found that the change in the granule sizes of all the three components did not significantly change the performance. The change in the amount of quartz wool for separation of the three layers also did not change the performance until the amount of quartz wool became too low to separate the catalysts completely. Thus, the most important conclusion or the key significance in our work is that the complete separation of the three catalyst components is a key to the selective formation of ethanol. This was already described in our manuscript (*please see Page 17, Lines 2-3*). We believe that this conclusion can offer important guidance for the future design of tandem catalytic systems for the synthesis of C₂₊ oxygenates from C1 molecules.

We have also analysed the reason behind this phenomenon. The key is that the oxygenate product, i.e., ethanol, is unstable once in contact with H-MOR, an acid catalyst for the carbonylation in the tandem system, which would easily catalyze the conversion of ethanol to ethylene. Thus, the separation of Pt-Sn/SiC and H-MOR-DA-12MR catalyst layers is necessary for obtaining high selectivity of ethanol. As requested by the reviewer, we have performed the conversion of ethanol over H-MOR-DA-12MR to provide direct evidence. We found that ethanol was completely converted with an ethylene yield of 97% under the reaction conditions similar to those used for syngas conversion (*please see Supplementary Table 14*). Moreover, we have also confirmed that the change in major product from ethanol to ethylene in the conversion of acetic acid, the intermediate product from syngas, when the amount of quartz wool for separation of H-MOR-DA-12MR and Pt-Sn/SiC became too low (*please see Supplementary Table 15*) or when the two components was in direct contact (*please see Supplementary Table 17*). All these results provide further evidence that the separation of catalytic components is required to avoid further conversion of ethanol. Based on these results and discussion, we have rewritten some sentences in the revised manuscript as follows: “*This suggests that the ethanol formed on Pt-Sn/SiC may undergo dehydration to ethylene on zeolite, which has direct contact with the Pt-Sn/SiC due to the too smaller amount of quartz wool. We confirmed that ethanol could be easily converted to ethylene over the H-MOR-DA-12MR under our reaction conditions (Supplementary Table 14). The formation of ethylene at the expense of ethanol has also been observed in the*

conversion of acetic acid upon increasing the proximity between the two components in H-MOR-DA-12MR | Pt-Sn/SiC (Supplementary Table 15)” (please see Page 16, Lines 9-15); “The conversion of acetic acid also led to the formation of lower olefins instead of ethanol over the mixed catalyst composed of Pt-Sn/SiC and H-MOR-DA-12MR in close contact (Supplementary Table 17).” (please see from Page 16-the last line to Page 17-Line 2).

Comment 4: There is a comment in the reviewer's response stating 'Regarding the reproducibility, our experimental results are reproducible and the relative deviation is less than 5% in all cases.' If the authors have already performed the experiments, it would be useful to include these data in the form of error bars or within the SI in a future revision.

Reply and actions taken: We appreciate this comment raised by the reviewer. Each of our catalytic reactions has typically been performed for three times and the relative deviation is within 5%. This confirms that our catalytic performances are reproducible. We have added the results containing errors bars in Figs. 2, 4 and 5 in the revised manuscript as well as in Supplementary Fig. 11 in the revised Supplementary Information. The following sentence has been added in the captions of these figures: “*The experiments in each case were performed for three times. The error bar represents the relative deviation, which is within 5%*”.

We have further added in the revised manuscript the reproducibility test result for triple tandem system ($K^+ZnO-ZrO_2$ | H-MOR-DA-12MR | Pt-Sn/SiC) for the conversion of syngas for six times with catalysts from different batches. We found that the performance with CO conversion of $4.0 \pm 0.2\%$ and ethanol selectivity of $81 \pm 4\%$ could be well reproduced (Supplementary Fig. 12). The following sentences have been added in the revised manuscript: “*The reaction with the triple tandem system was performed at 543 K for six times with catalyst from different batches, and the performance with CO conversion of $4.0 \pm 0.2\%$ and ethanol selectivity of $81 \pm 4\%$ could be well reproduced (Supplementary Fig. 12)*” (please see Page 12, Lines 3-5). The following sentence in the previous manuscript: “*The catalytic reactions were typically performed three times and the relative deviation was within 5% in each case*” has been revised into “*Each catalytic reaction was typically performed for three times and the relative deviation was within 5%, confirming that the catalytic performance is reproducible*” (please see Page 21, Paragraph 1, the last sentence).

General comment: Overall, the manuscript does present some interesting findings, but is not a good fit for Nature Communications.

Reply: As already mentioned above, different from previous papers that are devoted to the tandem catalysis for syngas conversions (including our previous paper published in *Angew. Chem.* 2018), the present manuscript reports our recent success in single-pass conversion of syngas to ethanol with high selectivity via triple tandem catalysis. Ethanol is not only an important bulk chemical but also an ideal fuel additive. The direct synthesis of ethanol from syngas is one of the most attractive but very challenging goals in catalysis. The selectivity of ethanol reported up to date typically cannot exceed ~60% even at limited CO conversions. The present work contributes to presenting a new triple tandem catalytic system, which achieves ethanol selectivities of 90% and 81% at CO conversions of 0.7% and 4.0%, respectively. At a CO conversion of ~10%, the ethanol selectivity can be sustained at 64%. Besides the careful design of efficient and selective catalyst for each step, we have also shown the crucial roles of the interplay between different reaction steps and the compatibility of catalysts in syngas stream. Our manuscript has demonstrated that the use of methanol carbonylation with CO in syngas is a promising strategy to control C–C coupling to form C₂ products with high selectivity. The encouraging results and new insights gained in this work would guide the design of efficient tandem catalytic systems not only for syngas but also for the transformation of other C₁ molecules (such as CO₂ and CH₄), and deepen our knowledge on the chemistry of tandem catalysis with interplay of multi-steps as well as compatibility of catalysts, thus providing a new opportunity to make breakthroughs in C₁ chemistry.

All the results and insights described above demonstrate once again that the tandem catalysis has become a very useful methodology in C₁ chemistry. It will certainly form a boom in the related research field (please see *Chem. Soc. Rev.* **48**, 3193-3228 (2019)), and we do not think that a couple of high-quality publications (like the *Angew. Chem. Paper*) are enough to address this booming field. Instead, we believe that there will appear a lot of high-quality work in the near future. Therefore, we think that our present manuscript is suitable for publication in *Nature Communications*.

Response to Reviewer 2

General comment: The author has made a more comprehensive introduction and appropriate comments on the progress of this field in their revised manuscript. They have also added more comparative experiments to prove the progress that they have made. In addition, more characterization evidence has been provided to support the structure of the catalysts. Therefore, in my opinion, this revised manuscript can be accepted.

Reply: We thank the reviewer for this kind evaluation on our manuscript.

Response to Reviewer 5

General comment: Multifunctional catalytic reactor operation is an important, clear new direction for bulk (petro)chemical industrial processes and hold promise for efficiency and energetic breakthroughs, eliminating intermediate separation processes and simplifying processing. Several examples have been given in literature for bifunctional systems, but in this manuscript three catalytic functions are combined in an operation where also thermodynamic constraints have to be faced, resulting in high product selectivity for an oxygenated product.

It is a new development compared with their former paper (ref 30), but with much lower selectivities and other products.

Here, the authors convincingly demonstrate the direct conversion of syngas mixtures into ethanol with high selectivity at a reasonable CO conversion in one reactor operation by a combination of three consecutive catalyst layers, each catalyzing cascade-wise methanol synthesis, methanol carbonylation and selective acetic acid hydrogenation. Mixing the catalysts resulting in more proximate/intimate catalytic functionalities did not work.

To operate in combination under similar conditions the three catalysts were successfully identified and optimized to be able to do so.

It must be realized for this case that the consecutive conversion process will still face the methanol synthesis thermodynamic limitation in the first step, i.e. limiting the CO conversion, but eliminates intermediate separation processing. On the other hand, the consecutive reaction requires a high CO/methanol ratio, so CO conversion cannot be too high to achieve a compatible operation, and a subtle balance between the performances of the catalysts had to be found in terms of catalyst properties, amounts and operating conditions. Limited syngas conversion is probably also the reason for the stable carbonylation performance where the hydrogen may remove or suppress coke deposition on the zeolite catalyst.

Of course, the limited CO conversion limits the yield of the ethanol product and separation of the reactor product mixture is still required. Some directions for further developments are mentioned.

The main manuscript is accompanied by extensive supporting information on catalyst characterization, composition and conditions dependencies, including new data. The role/function of the individual catalytic components is clearly demonstrated by the presented data.

Regarding this revision of the original manuscript, for which the authors have performed additional experimental work, they addressed the questions of the reviewers in a very adequate manner. Together with the various advanced

developments presented in this work (triple tandem operation for one-reactor syngas conversion into ethanol, subtle fine-tuned catalysts' optimization, compatible stable catalyst operation) I can only recommend acceptance of this manuscript.

Reply: We appreciate very much the positive evaluation by this reviewer.

REVIEWERS' COMMENTS:

Reviewer #1 (Remarks to the Author):

Thank you very much to the authors authors for doing an outstanding job addressing my meticulous comments, they have been informative and strengthened the manuscript. The manuscript can now be accepted for publication in Nature Communications.

Responses to Reviewers

Response to Reviewer 1

General comment: Thank you very much to the authors for doing an outstanding job addressing my meticulous comments, they have been informative and strengthened the manuscript. The manuscript can now be accepted for publication in Nature Communications.

Reply: We thank the reviewer for this kind evaluation on our revised manuscript. There are no revision requests now by the reviewer.